



# Cloud Phase Characteristics Over Southeast Asia from A-Train Satellite Observations

Yulan Hong and Larry Di Girolamo

Department of Atmospheric Sciences, University of Illinois at Urbana-Champaign, Urbana,
Illinois, USA

*Correspondence to: Yulan Hong (yulanh@illinois.edu)*

**Abstract.** This study examines the climatological characteristics of cloud phase over Southeast Asia (SEA) based on A-Train satellite observations. Using the combined CloudSat-CALIPSO (CC) data, five main cloud groups are investigated: ice-only, ice-above-liquid, liquid-only, ice-above-mixed and mixed-only clouds that have annual mean frequencies of 27.8%, 21.9%, 16.9%, 7.4% and 6.2%, respectively. Liquid-only clouds tend to occur in relatively cold, dry and stable environments. The other four cloud groups appear more frequently in relatively warm, humid and unstable conditions and their seasonal distributions move with large-scale climate systems: Asian monsoon and ITCZ. Liquid clouds are found to be highly inhomogeneous based on the heterogeneity index ($H_\sigma$) from Aqua MODIS, while ice-only and mixed-only clouds are often very smooth. Ice-above-liquid clouds are less inhomogeneous than liquid-only clouds due to the thin overlying ice clouds. Undetected subpixel liquid clouds enlarge clear sky $H_\sigma$, leading to some inhomogeneous clear sky pixels, which otherwise should be smooth. The reflectance at 0.645 ($R_{0.645}$) and brightness temperature at 11 μm ($BT_{11}$) of ice-only, liquid-only and ice-above-liquid clouds show peak frequencies near that of clear sky ($R_{0.645}$ ~ 0.02, $BT_{11}$ ~ 294 K), implying that a large population of thin or subpixel clouds over SEA are difficult to see by MODIS. The CC data also show abundant thin clouds over SEA, which indicates 78.3% ice-only and 89.6% ice-above-liquid clouds with ice $\tau < 3$. Mixed-only clouds in contrast are thick (top ~ 14 km), bright ($R_{0.645}$ ~ 0.61) and cold ($BT_{11}$~231K). The intraseasonal and interannual behaviors of all-sky $H_\sigma$ and spectral signatures follow that of cloud phases and are able to track the MJO and ENSO phases. Climatologies of clouds over SEA provided in this study are useful for the evaluation of large-scale model simulations, as well as for the interpretation of cloud properties derived from passive satellite sensors in this region.

## 1. Introduction

Cloud phases and their vertical structures are crucial to Earth's radiation budget (Hong et al., 2016; Li et al., 2011; Liou, 1986; Matus and L'Ecuyer, 2017; Oreopoulos et al., 2017), and insufficient knowledge in these areas has contributed to large uncertainties in current climate simulations. For instance, the simulated liquid and ice cloud amount and mass from global climate models (GCMs) show large discrepancies compared with observations, differing by orders of magnitudes in regions where clouds are ubiquitous such as the Western Pacific warm pool (Dolinar et al., 2014; Jiang et al., 2012; Kay et al., 2016; Waliser et al., 2009). The biases in modeled cloud properties are able to propagate and cause biases in other fields in the model, such as shortwave (SW) and longwave (LW) radiation (Li et al., 2013), sea surface temperature and precipitation (e.g., Grose et al., 2014). By examining cloud vertical structures, Cesana and Waliser (2016) found that most of the selected GCMs overestimate the frequencies of high-level clouds over tropical ocean and consistently underestimate low-level clouds. As a result, GCMs



produce insufficient heating near the surface and slightly over heating near the tropopause (Cesana et al., 2018). A recent study by Berry et al. (2019) showed that the Community Atmosphere Model, version 5 (CAM5) is in good agreement with observation in terms of ice cloud radiative effect, though CAM5 generates more frequent ice clouds than satellite
observations. These noted cloud and radiation biases ultimately point to an incomplete understanding of cloud phases, their vertical overlaps, and their interactions with large-scale circulations. Satellite observations continue to play important roles in furthering our understanding of cloud phases and vertical structures for future GCM evaluation.

       The radar on CloudSat and the lidar on Cloud-Aerosol Lidar and Infrared Pathfinder
Satellite Observations (CALIPSO) probing into atmosphere from space have offered unprecedented opportunities to explore cloud vertical details globally (Stephens et al., 2002; Winker et al., 2003). Using the combined CloudSat-CALIPSO (CC) observations, the vertical and horizonal structures of global hydrometeor have been first time shown in Mace et al. (2009). More details of cloud characteristics including their macrophyscial properties such as cloud
amount, heights and water mass, and microphysical properties such as effective radius (Re), and ice and liquid water content (IWC, LWC), have also been examined in many studies, including Eliasson et al. (2011) and Hong and Liu (2015) for ice clouds and Hu et al. (2010) for liquid clouds. With enhancement of our understanding of different cloud phase properties, these studies have assisted to improve GCM simulations (Kay et al., 2016; Zhao et al., 2018). The CC data
also helps characterizing cloud vertical overlaps, showing that multi-layer cloud occurrence frequency is greater than 50% in large-scale ascending regions such as the Western Pacific warm pool (Li et al., 2011; Matus and L'Ecuyer, 2017). Furthermore, Li et al. (2015) showed that cirrus, cumulus, altostratus and altocumulus tend to overlap with other cloud types. Oreopoulos et al. (2017) focused more on cloud altitudes by interpreting the overlap feature of high, middle
and low clouds, revealing that the two most prevalent cloud classes globally are single-layer low and high clouds (26% and 13.3%), respectively, followed by high over low clouds. Both Li et al. (2015) and Oreopoulos et al. (2017) showed distinct radiative effects between various cloud overlaps. Particularly, cloud radiative effects in the LW are reduced at the top of atmosphere (TOA) for high over low clouds compared to single-layer high clouds and much stronger than
single-layer low clouds, which nicely demonstrates the importance of accurately representing cloud vertical structures in GCMs. Although vertical structures for clouds have been examined in traditional designations, i.e. cloud types or cloud altitudes but not cloud phase, cloud phase has been demonstrated to be a sensitive parameter in GCMs that needs to be constrained (e.g., Cesana, 2016; Cesana and Storelvmo, 2017). Adding knowledge of cloud phases and their
overlaps will be beneficial to improve GCM performance.

       While cloud overlap or vertical heterogeneity is important in the radiative transfer, so is cloud horizontal heterogeneity (Marshak and Davis, 2005). It has long been demonstrated that the neglect of cloud horizontal heterogeneity with the plane-parallel assumption in radiative transfer can cause significant biases in computing irradiances and atmospheric heating rates (e.g.,
O'Hirok and Gautier, 2005), photolysis rates (e.g., Bouet et al., 2006), the emerging spectral and angular distribution of outgoing radiation field (Loeb and Davies, 1996) and in retrieving cloud microphysical properties from passive sensors (Loeb and Davies, 1996; Marshak et al., 2006). Many studies have since examined, at least in part, the global nature of these biases by successfully associating them with measured local spatial heterogeneity in SW radiance (Di
Girolamo et al., 2010; Ham et al., 2015; Liang et al., 2015). Zhang and Platnick (2011), for





instance, found that the differences of the Moderate Resolution Imaging Spectroradiometer (MODIS) liquid cloud Re retrieved at 3.7 and 2.1 μm increase with heterogeneity for inhomogeneous clouds. To avoid the difficulty in interpreting satellite products whose biases co-vary with scene heterogeneity, focus on examining the spatial-temporal variability of the raw
measured radiances in terms of their spatial heterogeneity and spectral signatures is a logical first step to understand Earth's climate systems – an approach that has successfully been carried out over the Terra satellite record over the globe (Zhao et al., 2016). However, the association of spectral and spatial heterogeneity signatures between different cloud phases and their overlap has yet been examined, which may be possible if cloud phase can be accurately characterized from
active space-based observations.

This study concentrates on cloud phases with an overall objective to investigate the characteristics of their climatology using the CC data with an emphasis on cloud phase overlap and the association with the spectral and spatial heterogeneity features from MODIS. We focus on South East Asia (SEA) because this region is strongly influenced by the Asian monsoon, the
intertropical convergence zone (ITCZ) (As-Syakur et al., 2016; Hong and Liu, 2015), the Madden-Julian oscillation (MJO) and the El Niño-Southern Oscillation (ENSO). All of them influence cloud systems and complicate cloud overlap structures. Also, current satellite products show wide ranging retrieval skills over SEA, in both aerosol and cloud properties, that are difficult to interpret (e.g., Reid et al., 2013). This has motivated several field campaigns in the
SEA environment to better characterize aerosol, cloud properties and their interactions, including 7-SEAS (7-Southeast Asian Studies) in 2010 and 2013 (Lin et al., 2013; Reid et al., 2013) and the CAMP2Ex (Cloud and Aerosol Monsoonal Processes Philippines Experiment) in 2019 (white paper in https://espo.nasa.gov/CAMP2Ex_White_Paper). Enhancing understanding of cloud phase characteristics can help interpreting satellite products and benefit future field
campaign preparation in this area. In addition, cloud phase climatology is needed to reduced the large cloud and radiation biases in GCMs that have been noted over SEA (Cesana et al., 2018; Waliser et al., 2009). By fusing the CC-MODIS data, this study explores the following scientific questions:

1.  What are the spatial patterns of cloud phases and their overlaps and how do these patterns
relate to large-scale dynamics?
2.  To what extent can the spectral and spatial heterogeneity signatures tell the features of cloud phases and their internal overlaps using the CC-MODIS observations?
3.  How do cloud phase characteristics vary in intraseasonal and interannual scales, i.e. their association with MJO and ENSO?

**2. Data and Methodology**

CloudSat, CALIPSO and Aqua are operated in the A-train satellite constellation, which is in a sun-synchronous orbit at an altitude of 705 km above Earth. The A-train's equator-crossing time is at around 1:30 pm in the daytime and 1:30 am at night. The tight orbit of these satellites allows the radar, lidar and MODIS to observe nearly the same point on Earth within 1 minute
(Stephens et al., 2018), and thus a straightforward match between these instruments can be performed. The information of all data used in this study has been summarized in Table 1. The SEA region is delineated with latitudes between 10°S and 30°N and longitudes between 80°-150°E.



## 2.1. The Combined CloudSat-CALIPSO Data

Launched in June 2006, the CloudSat satellite carries a cloud profiling radar operated at 94 GHz with a minimum sensitivity of -28 dBZ (Stephens et al., 2002, 2008). The radar's vertical resolution is 480 m but resampled to 240 m, while its horizontal resolution is 1.8 km along track by 1.4 km cross track. The radar is able to penetrate thick clouds but misses optically thin clouds and shallow clouds lower than 1 km altitude (Stephens et al., 2008). The lidar onboard CALIPSO launched in April 2006 has vertical and horizontal resolutions of 30 m and 333 m in the lower troposphere, and 60 m and 1 km in the upper troposphere (Winker et al., 2003). The lidar operates at 532 and 1064 nm and is suitable to detect optically thin clouds and aerosols, but its signal is easily attenuated, which limits its ability to penetrate optically thick clouds and to detect anything below. Nevertheless, the lidar has distinct advantages in detecting liquid clouds because 1) the backscattering ($\beta_c$) from water droplets is much less depolarized than that from ice particles and 2) water layers produce strong lidar returns that attenuate rapidly with altitude at cloud top (Hu et al., 2009; Wang and Sassen, 2001). In cases where thin clouds at any altitudes or shallow clouds near Earth's surface are missed by CloudSat, the CALIPSO lidar can detect these clouds if the lidar attenuation above these clouds is sufficiently small. Thus, combining CloudSat and CALIPSO is advantageous for detecting a wide range of cloud scenarios.

To utilize the complementary features of the CloudSat radar and the CALIPSO lidar, the CloudSat Data Processing Center provides a combined radar and lidar cloud classification product, known as 2B-CLDCLASS-LIDAR (Wang et al., 2012). This product reports cloud top and base heights for up to five layers. The cloud layer here extends from cloud top to its base, and the vertical space between two layers is more than 500 m (Sassen and Wang, 2008). Each cloud layer is assigned one thermodynamic phase, either liquid, ice or mixed. The 2B-CLDCLASS-LIDAR algorithm utilizes cloud top and base temperatures from reanalysis as a first cut for cloud phase determination. If cloud base temperature is lower than -38.5°C, this cloud layer is regarded as ice phase. Liquid phase is determined if cloud top and base temperatures are greater than 1° and -4°C, respectively. While in the temperature range (-40° - 0°C) where supercooled and mixed clouds would exist, potential liquid layers are first located using the feature of strong vertical gradient in lidar signals near liquid tops. If a liquid layer is detected by the lidar, the radar reflectivity factor (Ze) is further adopted to discriminate supercooled liquid from mixed cloud because Ze in mixed cloud is primarily contributed by ice particles. Temperature dependent Ze thresholds were generated to judge whether ice particles occur in the lidar detected liquid layer (Fig.2 in Zhang et al. (2010)). When the maximum Ze of the cloud layer is greater than the given threshold, the layer will be classified as mixed cloud, otherwise, it is classified as liquid cloud. Because of the combined sensitivity of the radar to thick clouds and the lidar to thin and liquid clouds, the vertical resolved cloud phase retrievals encompass a wide coverage of cloud thickness that range from subvisual cirrus to deep convective clouds. The cloud phase is also used to classify cloud types (e.g., cirrus, cumulus, stratocumulus etc.) within the 2B-CLDCLASS-LIDAR product (Wang et al., 2012), and used for downstream CloudSat product to retrieve cloud microphysical properties (Deng et al., 2010). The merged CC cloud phase product is suitable for the objective of this study to characterize the overlap features of all detected cloud phases. Four years of 2B-CLDCLASS-LIDAR data (2007-2010) are used.

The CloudSat level-2C ice cloud property product (2C-ICE) provides ice cloud optical (extinction coefficient in the visible) and microphysical (IWC and Re) properties retrieved from





combined CC measurements (Deng et al., 2010). The 2C-ICE algorithm first identifies ice clouds based on the cloud layer and phase from 2B-CLDCLASS-LIDAR. The vertical profiles of Re and IWC are retrieved based on an optimal estimation framework that minimizes a cost function linking the observed and estimated lidar backscatter and Ze via Gauss-Newton iteration. To
retrieve ice cloud properties, a modified gamma particle size distribution is adopted. To estimate lidar backscatter and Ze, parameterization of ice habits from Yang et al. (2000) is employed. The 2C-ICE product was evaluated in Deng et al. (2013) by comparing ice cloud properties to field campaign measurements. It was found that the flight-measured to 2C-ICE retrieved Re ratio is about 1.05 and the extinction coefficient ratio is about 1.03, hence suggesting excellent
consistency between 2C-ICE retrievals with in-situ observations. In this study, we integrate the retrieved extinction coefficient profile over depth of the ice layer to obtain ice cloud optical depth ($\tau$). 2C-ICE data from 2007-2010 is used.

       Meteorological data is also adopted to interpret the environment for different cloud phases. Temperature, wind and moisture are from the CloudSat's European Center for Medium-
Range Weather Forecasts auxiliary product (ECMWF-AUX, 2007-2010), which interpolates the ECMWF variables to each CloudSat profile (Cronk and Partain, 2017).

## 2.2. CC Profile Classification

       The CC profiles that contain clouds are classified into five groups (Fig. 1), according to the cloud layer and thermodynamic phase information from the 2B-CLDCLASS-LIDAR
product. The first group is ice-only cloud which refers to only ice phase identified in the profile. When ice layers occur above liquid layers, we classify the profile as ice-above-liquid cloud. Similarly, mixed-only, liquid-only and ice-above-mixed clouds are classified, respectively. We do not focus on liquid above ice, mixed above ice, mixed above liquid or liquid above mixed clouds, collectively referred to as 'other clouds', due to their low frequencies over SEA (~0.6%
in total over SEA). Note that one cloud phase in each cloud category contains both single and multiple layers. For example, for ice-above-liquid group, there could be more than one ice or liquid layers. This classification method simplifies the intricate cloud vertical structures but still catches the main features of cloud phase overlap.

## 2.3. The Aqua MODIS Data

The Aqua satellite, launched in May 2002, carries MODIS. MODIS has 36 discrete spectral bands, ranging from 0.415 to 14.235 μm, with spatial resolutions varying between 250 m to 1 km (Barnes et al., 1998; King et al., 1992; Platnick et al., 2003). To allow collocating CC and MODIS pixels, the MYD03 product, which includes longitude, latitude, solar zenith angle and land/sea mask, is used to obtain MODIS geolocation information. The nearest MODIS 1 km
resolution pixels are assigned to the CC data from 2007-2010. The distance of the collocated CC-MODIS pixels is usually smaller than 700 m. No other resampling step is performed. Differences in the sensitivities and instantaneous fields of view of the instruments are kept in mind during the analysis.

       To investigate the spectral signatures of different cloud phases, the MODIS Collection
6.1 level 1B calibrated radiance data (MYD021KM) is used. The bands selected in this research have center wavelengths at 0.645, 1.375, 1.64, 2.13, 8.55 and 11.03 μm, which are commonly used to distinguish cloud phase from space (e.g., Marchant et al., 2016). Ice crystals are more absorptive at the shortwave infrared (SWIR) (e.g., 1.64, 2.13 μm) than liquid droplets and thus



ice clouds have smaller reflectance (R) at TOA; whereas ice and liquid clouds are about equal in R at the visible (0.645 μm) for the same cloud optical depth and particle size, which forms the basis for cloud optical and microphysical retrievals and cloud phase classification for MODIS (Marchant et al., 2016). $R_{1.375}$ depends on cloud optical thickness and the amount of water vapor above the cloud, since 1.375 μm lies at the center of a strong water vapor absorption band. If cloud top is at a low altitude, the solar photons at 1.375 μm will be largely absorbed by water vapor above cloud leading to near-zero $R_{1.375}$ (Marchant et al., 2016). In the IR, we convert the radiances of 8.55 and 11.03 μm to brightness temperature (BT). $BT_{8.5}$, $BT_{11}$ and the difference between $BT_{8.5}$ and $BT_{11}$ (BTD) are sensitive to cloud top temperature, thickness and phases (Baum et al., 2012).

The MODIS level 2 cloud product (MYD06) provides cloud phase information identified according to three different IR channel pairs, i.e. 8.5 and 11 μm, 11 and 12 μm, and 7.3 and 11 μm, known as the IR-only algorithm (Baum et al., 2012). The product contains three cloud phases: ice, water and undetermined (see details in Baum et al. (2012)). Cho et al. (2009) evaluated MODIS IR-only cloud phase using CALIPSO observations. They found that agreements of MODIS to the CALIPSO top layers are 64% and 34.7% respectively for CALIPSO detected liquid and ice clouds over the globe. We revisited the comparisons of the latest version of MODIS IR-only cloud (C6.1, complete on Mar. 2018) to the CC cloud phase (Fig. 1) over SEA in a similar way as Cho et al. (2009). About 65% CC liquid-only clouds agree with MODIS liquid clouds, which is similar to the global results of Cho et al. (2009). About 62% of CC ice-only clouds are reported to be ice by MODIS over SEA, agreeing better than the global results of Cho et al. (2009). In addition, most of CC ice-above-liquid (56%), ice-above-mixed (74%) and mixed-only clouds (71%) are reported to be ice phase by MODIS. We also found that 91% of CC clear sky is reported as clear sky by MODIS, indicating that MODIS misses some thin cirrus in the SEA region (Reid et al., 2013). More details about the CC and MODIS cloud phase comparison is displayed in Table 2. In this study, MODIS cloud phase is also adopted to obtain additional cloud phase properties because of MODIS' wide viewing swath (2330 km) and its longer time period than the CC data.

The spatial heterogeneity index ($H_\sigma$) defined as standard deviation over the mean of measured radiances of sixteen 250 m pixels within a 1-km sample (Liang et al., 2009) is also included in the MYD06 product. The $H_\sigma$ usually increases with subpixel-level inhomogeneity and correlates with radiation and remote sensing biases rooted in plane-parallel assumption (Cho et al., 2015). $H_\sigma$ is reported at 0.645 and 0.865 μm. Here, we adopt $H_\sigma$ at 0.645 μm because $H_\sigma$ for 0.865 μm is reported to be zero for saturated pixels, which occurs for thick clouds under certain sun-view geometries encountered in the MODIS data.

The matched $H_\sigma$ values and MODIS radiances are assigned to cloud phase from 2B-CLDCLASS-LIDAR to investigate cloud spatial heterogeneity and spectral radiation features. Longer MODIS data (2003-2017) is used for analysis of interannual variations of cloud phase. Considering that no visible and SWIR radiances are available at night, only daytime data is considered through the paper.

## 2.4. Meteorological indices

The MJO (Madden and Julian, 1971) consists of large-scale coupled atmospheric circulation and deep convection in the tropical atmosphere. It forms in Indian Ocean and





propagates eastward at a speed around 5 m s⁻¹ across the Maritime Continent and into the
equatorial western/central Pacific oceans with an intraseasonal variability of 30-90 days (Zhang,
2005). To understand cloud phase evolution with the MJO, we adopt the Real-time Multivariate
MJO (RMM) index (Wheeler and Hendon, 2004), which defines eight MJO phases using two
leading Empirical Orthogonal Functions (EOFs) of combined 850, 200 hPa zonal wind from
NCEP reanalysis and satellite-observed outgoing longwave radiation (OLR) over the tropical
belt. We only focus on strong MJO events with amplitude greater than one. The RMM index is
available at http://www.bom.gov.au/climate/mjo/.

The ENSO has a interannual variability of 3-5 years, and the multivariate ENSO index
(MEI) is well suited to identify ENSO events (Wolter and Timlin, 1993, 1998). The new version
of MEI is created by the EOF analysis of five variables including sea level pressure, sea surface
temperature, surface zonal and meridional winds, and OLR. The ENSO index is available at
https://www.esrl.noaa.gov/psd/enso/mei/.

## 3. Results

### 3.1. Seasonal Variations

### 3.1.1 Meteorological Conditions

To better understand the linkages between cloud properties and large-scale dynamics,
meteorological fields are presented in Fig. 2. It shows temperature, specific humidity, wind field,
and static stability at the lower (~ 850 hPa) and upper troposphere (~ 180 hPa) over SEA in four
seasons, that is, boreal spring (March, April and May (MAM)), summer (June, July, and August
(JJA)), autumn (September, October and November (SON)) and winter (December, January and
February (DJF)).

In the lower troposphere, relatively high and homogeneous temperatures are observed all
year around (~290 K) corresponding to the Indo-Pacific warm pool. However, temperatures drop
in boreal winter (< 285 K, Fig. 2a4) and raise in summer (> 290 K, Fig. 2a2) over Southeast
China and South China Sea. In the upper troposphere, temperatures are relatively low at latitudes
between 10°S-10°N (< 213 K), while temperatures over South Asia in summer are 1-3 K higher
than in other seasons. Similarly, high humidity (> 10 g kg⁻¹ at lower troposphere and > 0.03 g kg⁻¹
at upper troposphere) is located at South of 10°N latitude during spring and winter (Figs. 2b1,
b4), while in autumn (Fig. 2b3), the humidity pattern is quite symmetric to the equator. Also, air
is especially moist over South Asia during summer than other seasons at both lower and upper
troposphere. The summer high temperature and humidity over South Asia are related to the
heating and convection over Tibetan Plateau which maintain a hot and humid upper troposphere
and the South Asian Anticyclone (Yeh, 1982). The summer monsoon also helps transfer a large
amount of moisture from the Indian ocean to Asia (Figs. 2c2).

The seasonality of the wind field is evident (Figs. 2c1-c4). In the lower troposphere, the
southwesterly wind flow brings warm and humid air to South Asia in summer when ITCZ is
located at the North of equator, providing favorable conditions to form clouds and precipitation
(summer monsoon). The wind direction at upper troposphere is northeasterly which is nearly
opposite to that at the lower troposphere. In DJF, ITCZ shifts to southern hemisphere and the
wind flow reverses. The prevailing northeasterly flow near the lower troposphere (winter
monsoon) is also opposite to the wind direction at the upper troposphere (southwesterly).





However, the upper troposphere wind is much weaker in winter than in summer because the summer South Asian Anticyclone above Tibetan Plateau enhances the upper troposphere wind flow (Yeh, 1982). The spring and autumn are two transition seasons of summer and winter monsoonal flows.

The lower-troposphere static stability (LTSS) is estimated as the difference of potential temperature between surface and around 3 km above and the upper-troposphere static stability (UTSS) is the potential temperature difference between the tropopause height and 3 km below. The tropopause height is defined following the World Meteorological Organization, i.e. the lowest level where lapse rate is $2°$ C km$^{-1}$ or less and the average lapse rate between this level

and all higher levels within 2 km is smaller than $2°$ C km$^{-1}$ (Grise et al., 2010). Figures 2d1-d4 reveal that LTSS is usually smaller over land than over ocean. Small LTSS values ($< 4$ K km$^{1}$, yellow-green color) over ocean correspond to a wetter atmosphere (Fig. 2b). Relatively larger LTSS ($> 4$ K km$^{-1}$) occurs in winter and spring such as over East and South China sea. The LTSS has been proven to be an important parameter indicating low-level cloud formations. For

instance, Klein and Hartmann (1993) showed that $1°C$ increase in stability associates with a 6% increase in stratus cloud area coverage. The pattern of UTSS is similar with that of LTSS.

### 3.1.2 Occurrence of All Clouds

This section focuses on cloud spatial distributions over SEA. The horizontal occurrence frequency, defined as the ratio of total cloudy number to the total observation sample in each 5°

long x 2° lat grid derived from the 2B-CLDCLASS-LIDAR data, is shown in the upper panels of Fig. 3. The latitude-altitude cross sections centered at 115°E, obtained by cloudy number in each 5° long x 2° lat x 250 m height cell divided by the observation sample in that cell, are displayed in the lower panels of Fig. 3.

Over SEA, the annual mean cloud frequency is about 80.9% being the smallest in winter

and the largest in summer (Table 3). As expected, Large cloud occurrence frequency associate with large humidity, warm temperature and low stability in the lower and upper troposphere (Figs. 2, 3). For instance, the cloud occurrence frequency in Indochina is nearly 100% during the summer monsoon season, while in winter when the monsoon has retreated, these regions have much lower cloud occurrence (30-40%). The cloud pattern accordingly moves to the Malaysia

and Indonesia regions as the ITCZ shifts southward. However, not all clouds favor a warm, humid and unstable condition. Frequent cloud occurrence (~70%) is also observed over Southeast China and East China Sea during winter when the atmosphere is cold, dry and stable (Figs. 2, 3a4). These clouds usually have low cloud heights ($< 5$ km) as seen from the cross section (Fig. 3b4). The cloud pattern and its seasonal variation derived from the CC observations

largely agree with those reported over SEA derived from MISR, MODIS and CALIPSO, although the mean value of cloud occurrence varies among different platform due to different instrument sensitivity (Reid et al., 2013).

The cross sections centered at 115°E (Figs. 3b1-b4) display prevailing high-level clouds located at around 10-15 km, matching to the ubiquitous cirrus clouds over the Warm pool

regions (Sassen et al., 2008). These ice clouds occur north of 10°N in summer but shift to the South in winter, i.e. moving with the ITCZ and monsoon climate systems. Also, low-level cloud (e.g. $< 3$ km) frequency can exceed 30% near the boundary and generally match to the boundary clouds derived from MISR (Reid et al., 2013), owing to MISR's ability to detect the cloud top





heights of low clouds in the presence of cirrus. Low-level clouds cloud have a high chance to be covered by the upper ubiquitous ice clouds (Yuan and Oreopoulos, 2013), which is further quantified in next section.

### 3.1.3 Occurrence of Cloud Phases

5         Figure 4 shows horizontal and vertical distributions of the five cloud groups as defined in Fig. 1. The mean occurrence frequency of each cloud class in four seasons over ocean and land is summarized in Table 3. The five cloud groups display visible differences in both of their mean frequencies and spatial distributions. Ice-only clouds (Figs. 4a1-a4) occur the most frequent (~ 27.8%) among all cloud classes with much higher frequency over ocean (~31.3%) than over land
(~18.4%). These clouds appear more frequently in summer (~34.7% over ocean, ~19.6% over land) and prefer the locations at north of the equator but move to the south in winter leading to a smaller mean frequency over SEA (~26.5% over ocean, ~16.4% over land). Ice-only clouds mainly locate at high altitudes between 10-15 km corresponding to the tropical cirrus (Hong and Liu, 2015; Reid et al., 2013; Sassen et al., 2008).

15         Liquid-only clouds (Figs. 4b1-b4) have an annual mean frequency of ~16.9%. They are widely distributed over Southeast China and East China Sea which increase in fall, reach to a maximum in winter (> 50%) and remain ubiquitous until spring. These clouds are the so called 'Chinese stratus' by Klein and Hartmann (1993) associated with lower-troposphere cold and dry air and large LTSS (Fig. 2). However, liquid-only clouds have very small frequencies (< 10%)
between 10°S-10°N where widely distribute ice clouds, which indicates that liquid clouds occurring here are likely being covered by ice clouds, hence, they are grouped as ice-above-liquid cloud class.

        Ice-above-liquid clouds have an annual mean occurrence frequency of ~ 21.9% (Figs. 4d1-d4). Ice layers located at 10-15 km cover the underlying liquid clouds mostly with height
below 3 km. These clouds occur frequently over South China and Indochina during the summer monsoon season (Fig. 4d2), with the mean frequency over land (~35.3%) being much larger than over ocean (22.1%) (Table 3). In winter, ice-above-liquid clouds move to West Pacific Ocean and Malaysia-Indonesia regions (Fig. 4d4).

        Mixed phase related clouds, i.e. mixed-only (annual frequency ~ 6.2%) (Figs. 4c1-c4)
and ice-above-mixed clouds (annual frequency ~7.4%) (Figs. 4e1-e4) have a higher frequency over land than ocean (Table 3), which is consistent with more convective activities over land in the afternoon (Eastman and Warren, 2013). The mixed-only clouds are mature convective clouds as seen from their cross sections which extend from near surface up to above 15 km. The frequency of this cloud group agrees with that of precipitation (Adler et al., 2001). Some mixed-
only clouds also appear near 30°N consistent with the location of liquid-only clouds but their tops being much higher than liquid-only clouds. The ice-above-mixed clouds are more likely under development with mixed layer tops reaching to around 10 km. If the mixed layers develop higher, they would merge with the overlying ice clouds and are classified into mixed-only cloud class. Both mixed-only and ice-above-mixed clouds show relatively large frequencies near the
west coast of Indochina in summer and over Malaysia-Indonesia in winter that relate to the monsoon and ITCZ as well as topographical effect.

        Overall, liquid-only clouds occur associated with high LTSS and lower temperature in the lower troposphere, agreeing with the relationship of low-level cloud with stability (Klein and



Hartmann, 1993; Li et al., 2014). In contrast, ice-only, ice-above-liquid, ice-above-mixed and mixed-only clouds, collectively named as 'ice-contained clouds', favor a humid, warm and unstable environment.

Figure 5 further summaries the mean and standard deviation of the meteorological variables discussed in Fig. 2 from the ECMWF Aux product for the five cloud groups in summer and winter seasons over ocean (to avoid the low static stability over land). In the lower troposphere, all cloud groups in summer tend to have smaller LTSS, higher temperature and humidity than in winter (Figs. 5a, b). The standard deviations in summer are also smaller, being consistent with a more homogeneous spatial pattern of the meteorological fields (Fig. 2). In

winter the liquid-only clouds tend to have a much smaller humidity and colder temperature corresponding to the occurrence of the 'Chinese stratus' (Klein and Hartmann, 1993). For those 'ice-contained clouds', they are still located in a relatively warm, moist and unstable atmosphere but their standard deviations are much larger than that in summer, agreeing with the less homogeneous spatial pattern of meteorology in winter (Fig. 2). In the upper troposphere, the

relationship between the five cloud types and meteorology is similar in both summer and winter with liquid-only clouds deviating from 'ice-contained clouds'. The 'ice-contained clouds' relate to smaller UTSS as reported in Li et al., (2014). Also, ice-only and ice-above-liquid clouds share very similar upper tropospheric meteorology as their mean and standard deviations are nearly the same, which is not surprising because the low stability and high moisture are essential to

maintain cirrus (Christensen et al., 2013; Li et al., 2014). However, the specific humidity in the upper troposphere of ice-above-mixed clouds is larger than both ice-only and ice-above-liquid clouds (Figs. 5c, d), which may reveal that convection below brings moisture to the upper troposphere.

### 3.1.4 Distributions of Cloud Phase Properties

Figure 6 presents the probability distribution function (PDF) of cloud properties including cloud tops for all cloud phases, and base, geometric thickness, τ and Re for ice layers. The averages of the cloud properties in the four seasons are summarized in Table 4.

We first focus on the properties of ice layers in the three categories: ice-only, ice-above-liquid and ice-above-mixed clouds (Figs. 6a-e). We combine both land and ocean data to

investigate the distributions of ice layer properties, because their PDFs display similar shapes between that over land and ocean (Figure not shown), yet their averages are separately summarized in Table 4. The three categories of ice layers share many similarities in their PDFs. The modes of ice top PDFs (~ 16 km) and base (~ 12.5 km) are slightly greater than their means and medians (Fig. 6 and Table 4). The samples with ice top > 15 km or ice base > 10 km account

for more than 60% and 70%, respectively. The modes of geometrical thickness PDFs are around 1 km which are also smaller than their mean and median values of 2-4 km (Fig. 6c and Table 4). In addition, more than 60% samples have geometrical thicknesses less than 3 km. These statistics demonstrate that the distributions of ice clouds skew to higher locations and thinner thickness, corresponding well to the cirrus near tropopause (Haladay and Stephens, 2009; McFarquhar et

al., 2000). The τ PDFs show two modes for the three types of ice layers, and the Re PDFs with their modes slightly greater than 15 μm. Also, the ice layers tend to be thicker, i.e. larger geometrical and optical thickness, with larger ice particles in summer and autumn than in the other two seasons (Table 4).




There also exist differences between the three groups of ice layers. For example, ice-only clouds tend to locate 0.4-1.6 km lower over land than ocean, and the lower location may allow more moisture to feed into ice clouds, which may explain the reason that ice clouds over land have mean τ values of 1-2 and Re of 3-5 μm larger than those over ocean (Table 4). Compared to ice-only clouds, the ice layers above liquid or mixed clouds show much less land-ocean contrast in these properties (Table 4). These two groups of ice layers have similar medians and means, and contain about 70% samples with geometrical thickness < 3.0 km and about 90% samples with ice τ < 3.0--the τ threshold of cirrus (Sassen et al., 2008) (Figs. 6 c, d). In contrast, ice-only clouds are thicker with larger means, medians, Re (Table 4) and less samples with τ < 3.0. This is contributed by more ice-only cloud samples with geometrical thickness > 3.5 km, τ > 1.6 and Re > 60 μm (Fig. 6). It should be noted that the CALIPSO lidar signals will be totally attenuated by thick ice clouds (τ > 3.0) (Sassen et al., 2008) and the CloudSat radar fails in detecting shallow cumulus. Due to these instrument limitations, some ice-above-liquid clouds (e.g., thick ice over shallow liquid) could be classified into ice-only group, leading to some sampling biases to the mean ice τ and Re. However, in the ice τ range of 1.6-3.0 where CALIPSO can penetrate the cloud, ice-only clouds have a total frequency of ~ 0.2 in this τ range, being higher than the frequency (~ 0.14) of the other two groups and demonstrating a higher probability of ice-only clouds being thicker.

For liquid or mixed layers, we only focus on cloud tops, because the determination of cloud base suffers from larger uncertainties than cloud top due to the limitation of instruments, i.e. CloudSat radar has difficulty in distinguishing the cloud base near Earth surface and the CLIPASO lidar signal is easily attenuated by liquid/thick clouds (Hu et al., 2009; Stephens et al., 2008). For the liquid top PDFs (Fig. 6f), there are two modes for liquid below ice clouds (green). One mode is located at 1 km for ocean (2-3 km for land), agreeing with results from MISR (Reid et al., 2013), and the other is located at ~ 6 km for both ocean and land. We further obtain the spatial distributions of liquid below ice clouds with liquid top greater and lower than 5 km, respectively (Figure not shown). It is shown that liquid clouds with top < 5 km are widely distributed over SEA, while those with liquid top > 5 km are more concentrated in locations with latitude < 10°, corresponding to a more unstable environment as shown in Fig. 2. In another word, larger moisture and small LTSS allow liquid clouds to develop deeper. For liquid-only clouds, they have a much higher frequency than those liquid below ice clouds at the PDF mode of 1 km, and the second mode is not evident for liquid-only clouds. Since the frequencies at top > 5 km of liquid below ice are much larger than liquid-only clouds, the averaged liquid top value of the former is higher than the latter (Table 4). The top difference is much larger in summer and fall (~ 1 km over ocean and 300 m over land).

For the mixed layers below ice clouds (Fig. 6g), the mode of cloud top PDF (cyan) is at around 6 km and the mean is about 9 km with the values over ocean about several hundred meters higher than over land (Table 4). Note that as the mixed layers develop deeper, e.g. top > 10 km, and merge with the upper ice layers, these clouds would be grouped as mixed-only clouds. For mixed-only clouds, the primary mode of cloud top PDF is at around 16 km and the secondary mode is at around 6 km. The primary mode is much higher over ocean than over land, while the secondary mode is higher over land, indicating that more mixed clouds over land are under development around 1:30 pm local time. This agrees with the results in Nesbitt and Zipser (2003) that convective clouds keep developing in the early afternoon and reaching to intensity





maximum in the late afternoon over land, while diurnal variation of convection intensity is insignificant over ocean.

### 3.1.5 Spatial Heterogeneity of Cloud Phase

Sections 3.1.1 – 3.1.4 have primarily displayed the macrophysical properties including
spatial distributions, cloud thickness, top and base heights for the different cloud phases and their relationship with meteorology. In this section, by combining the MODIS and 2B-CLDCLASS-LIDAR data, we investigate cloud spatial heterogeneity ($H_\sigma$), which is not only closely related to cloud micro and macrophysical properties but also affects the accuracy of cloud retrievals from passive sensors and radiative transfer modeling (Ham et al., 2015; Zhang and Platnick, 2011).
Only ocean data is considered to avoid complications with the effects of land surface heterogeneity on interpreting results.

The $H_\sigma$ PDFs for the CC clear sky and the five cloud groups are shown in Fig. 7a. As displayed, the PDF of CC clear sky has a sharp peak at $H_\sigma \sim 0.01$, suggesting that clear sky is usually spatial homogeneous. For cloudy sky, liquid-only clouds are the most heterogeneous
among all cloud groups which have $H_\sigma$ ranging from 0.01 to 1 with a peak located at $H_\sigma \sim 0.5$. The ice-only clouds in contrast are homogeneous as the PDF has a peak close to that of clear sky. This suggests that the biases in retrieved optical and microphysical properties of clouds from passive sensors caused by the plane-parallel assumption will be larger for water clouds compared to ice clouds in the SEA region. Indeed, MODIS liquid Re difference retrieved from three
wavelengths (1.6, 2.1 and 3.7 μm) is especially large over this region (up to 10 μm) (Zhang and Platnick, 2011), and hence, it needs to be careful when interpreting the cloud-aerosol relationship over SEA using the liquid cloud Re from passive sensors (e.g., Ross et al., 2018) considering that the large spatial heterogeneity of liquid clouds would contribute large biases to the retrieved Re. For ice-above-liquid clouds, the PDF curve moves slightly to a smaller $H_\sigma$ region compared to
liquid-only clouds as the overlying ice clouds have a spatial smoothing effect on the radiation emerging from the liquid clouds below. However, due to the thin features of overlying ice clouds, radiation from underlying liquid clouds dominate (Sect. 3.1.4). For this reason, MISR's stereoscopic technique, which matches spatial patterns between two views for retrieving cloud top height, often retrieves the height of the lower liquid layer rather than the height of the thin ice
layer (Reid et al., 2013; Stubenrauch et al., 2013). The $H_\sigma$ PDF of ice-above-mixed clouds is similar with that of ice-only clouds but with some samples having $H_\sigma$ smaller than clear sky (e.g., $H_\sigma < 0.01$). This feature is more obvious for the mixed-only clouds that have about 50% samples with $H_\sigma < 0.01$ (Fig. 7b), indicating that these clouds are extremely homogeneous. We aware that these cases of very smooth ice-above-mixed and mixed-only clouds correspond to
high mixed layer tops and large reflectance at 0.645 μm (discussed in next section), revealing the situations associated with deep convection. While these clouds are locally homogenous, hence favoring the plane-parallel assumption in radiation computation (Ham et al., 2015).

Note that CC clear sky, ice-only, ice-above-mixed and mixed-only clouds are usually homogeneous, but there exist some heterogeneous cases, particularly for clear sky whose PDF
has a long tail extending to large $H_\sigma$ values up to 1 (Fig. 7a) and has about 20% samples with $H_\sigma$ values greater than 0.1 (Fig. 7b). Mismatch of pixels in collocation or difference in spatial resolutions of CC (1.8 km x 1.4 km) and MODIS (1 km) can contribute uncertainties to the $H_\sigma$ of CC clear sky. Yet with a focus on the pixels reported to be clear sky by both CC and MODIS (not shown), they behave nearly same as that of CC clear sky as shown in Fig. 7a, which stands



to the reason that the long tail is due to the significant amount of misdetection of small subpixel clouds by both MODIS and CloudSat. Indeed, many small liquid clouds with size ranging in a few tens to hundreds of meters (e.g., Koren et al., 2008) that are difficult to be measured by MODIS as reported in Zhao and Di Girolamo (2006). Also, applying the Advanced Space

Thermal Emission and Reflection Radiometer (ASTER) data (15 m resolution) to exclude the MODIS clear sky pixels that contain ASTER reported clouds significantly reduce the long tail of $H_\sigma$ PDF but with its mode unchanged (consult with Guangyu Zhao), which further validates the undetected clouds in some MODIS clear sky pixels and the typical clear sky $H_\sigma$ value of ~0.01.

Moreover, we calculate the MODIS liquid cloud fraction, defined as the ratio of liquid

cloud samples based on the MYD06 product to the total 25 pixels in a 5 km by 5 km surrounding of the collocated CC-MODIS pixel. As shown in Fig. 7c, larger fraction of MODIS liquid clouds is observed around the CC clear sky pixels as $H_\sigma$ increases. The ubiquitous liquid clouds in the surrounding indicate a high chance of being polluted by small liquid clouds in the subpixels. Similarly, the MODIS liquid cloud fractions in the surroundings of CC ice-only, mixed-only and

ice-above-mixed clouds increase with $H_\sigma$ values as well, demonstrating that the heterogeneous cases of these clouds could also be due to undetected liquid clouds in the subpixel.

Figure 8 displays the spatial distributions of $H_\sigma$ in four seasons for CC clear sky, the five cloud groups and all sky that includes both clear and cloudy sky. As expected, ice-only, ice-above-mixed, and mixed-only clouds are homogeneous everywhere ($H_\sigma < ~ 0.05$) with some

relatively large values ($H_\sigma ~ 0.1$) in the liquid-only cloud prevailing regions such as East China sea in winter (Figs. 4, 8b4, 8d4, 8f4). As liquid-only clouds are the most heterogenous, they show the largest spatial $H_\sigma$ values over SEA (Figs. 8c1-c4). Also, the $H_\sigma$ values over East China sea in spring, fall and winter are relatively smaller than the $H_\sigma$ in other regions, implying that the 'Chinese stratus' named by (Klein and Hartmann, 1993) that favor a dry and stable

meteorological conditions are less heterogenous than other liquid-only clouds (Figs. 8c1-c4). Ice-above-liquid clouds (Figs. 8e1-e4) are smoother than liquid-only clouds and relatively small values tend to coincide with frequent ice-above-liquid cloud occurrence frequency that associates with monsoon and ITCZ such as in Indonesia-Malaysia region in winter or North Indian Ocean in summer (Fig. 4). The $H_\sigma$ pattern of CC clear sky (Figs. 8a1-a4) displays smaller values than

ice-above-liquid clouds but larger than ice-only, ice-above-mixed and mixed-only clouds. It is because $H_\sigma$ PDF of clear sky has higher frequency than the other three cloud groups when $H_\sigma >$ 0.2 (Fig. 7a). Also, the places with large $H_\sigma$ values of CC clear sky are consistent with those of ice-above-liquid or liquid-only clouds, which in turn proves the high chance of liquid clouds increasing the subpixel variability.

For all sky (Figs. 8g1-g4), the small $H_\sigma$ values occur in North of the equator in summer, including Indian Ocean and South China sea, and the pattern is quite symmetric to the equator in fall and moves to South of the equator in winter, being consistent with the shift of climate system, i.e. Monsoon and ITCZ. It is because the small $H_\sigma$ values are primarily contributed by ice-only clouds due to their large occurrence frequency (Fig. 4) and spatial homogeneous

features. In contrast, the pattern of large $H_\sigma$ agrees more with that of liquid-only cloud occurrence (Fig. 4).

Overall, liquid clouds are spatial heterogeneous over SEA, whereas ice-only and mixed clouds are usually homogeneous. Ice-above-liquid clouds are less heterogenous than liquid-only clouds due to the smoothness of the overlying ice clouds, but their $H_\sigma$ values are still large



because overlying ice clouds are thin and the emerging radiance from underlying liquid clouds dominates. Clear sky is usually smooth but undetected liquid clouds increase it subpixel variability. The seasonal variations of all-sky $H_\sigma$ spatial patterns are in accordance with cloud movements that associate with monsoon and ITCZ.

### 3.1.6 Spectral Radiative Features

The spectral differences in the complex refraction index between water and ice, as well as the differences between the macro and microphyscial properties of ice and liquid clouds, lead to differences between cloud phases in the observed spectral radiance reaching to the satellite sensor. This section examines the spectral radiance at the TOA observed by MODIS for the CC clear sky and the five cloud groups defined in Fig. 1 to investigate their radiative features. Similar to Sect. 3.1.5, only ocean data is adopted. The averages of the reflectance (R) and the brightness temperature (BT) for each cloud group are shown over SEA in Fig. 9, while the PDFs and the cumulative distribution functions (CDFs) are displayed in Fig. 10.

At 0.645 μm, the averaged reflectance for clear sky is about 0.04 (Fig. 9) and the PDF shows a narrow peak at $R_{0.645} \sim 0.02$ (Fig. 10a1). The average $R_{0.645}$ of ice-only cloud is about 0.14. The location of its PDF peak is nearly same as clear sky and its CDF shows that about 80% samples with $R_{0.645} < 0.2$, proving the thin features of ice-only clouds over SEA and agreeing with their small optical depths in Fig. 6d. Similarly, for liquid-only cloud, the PDF also shows its mode nearly that of clear sky. A large fraction of liquid-only clouds that are optically thin clouds (e.g., more than 75% of liquid-only clouds with $R_{0.645} < 0.2$), being consistent with the findings of Leahy et al. (2012) that thin marine low-cloud fraction is greater than 80% for the SEA region. Ice-above-liquid clouds have an average $R_{0.645}$ of 0.21. The peak of its PDF located at $R_{0.645} \sim 0.03$ is lower than that of ice-only and liquid-only clouds, and the corresponding $R_{0.645}$ CDF is towards to a larger reflectance region, demonstrating that ice-above-liquid clouds in the column are thicker than either ice-only or liquid-only clouds. But ice-above-liquid clouds still contains more than 60% samples with $R_{0.645} < 0.2$, further demonstrating the ubiquity of thin clouds over SEA. Thin clouds would be difficult to see by MODIS as shown in Table 2 that around 10% of CC ice-above-liquid cloud, 30% of CC liquid-only and ice-only cloudy samples are reported to be MODIS clear sky. The largest average $R_{0.645}$ is seen in mixed-only clouds ($\sim 0.61$), followed by the ice-above-mixed clouds ($\sim 0.46$). Their PDFs are broad and flat, but with the frequencies at $R_{0.645} > 0.4$ are evident in Fig. 10a1. Their CDFs reveal that 60% of ice-above-mixed and 80% of mixed-only clouds have $R_{0.645} > 0.4$, indicating that these clouds are geometrically deep (consistent with Fig. 4) and optically thick.

At 1.64 μm , the average $R_{1.64}$ of ice-above-liquid clouds is nearly same as liquid-only clouds ($\sim 0.13$) (Fig. 9). Both these cloud groups are obviously more reflected than ice-only clouds (average $R_{1.64} \sim 0.06$). The average $R_{1.64}$ of ice-above-mixed and mixed-only clouds is about 0.19, implying that ice-above-mixed clouds are optically thick enough to reach the asymptotic reflectance and mixed-only clouds that are thicker do not increase $R_{1.64}$. Similarly, at 2.13 μm, the average $R_{2.13}$ of both mixed-only and ice-above-mixed clouds is also nearly equal (0.11), which in turn demonstrates their large thickness. The curve of $R_{2.13}$ across different cloud groups shows a similar shape with that of $R_{1.64}$ but with a smaller magnitude owing to the larger imaginary part of reflective index of water and ice at 2.13 μm compared to 1.64 μm. Particularly, the imaginary part of ice reflective index is larger than water at both wavelengths. Hence, when examining the reflectance ratio of SWIR (1.64 or 2.13 μm) to the visible (0.645 μm), we would



expect that liquid-only clouds show a larger ratio than any other cloud groups. Because the Aqua MODIS 1.6 µm band has many dead detectors (King et al., 2013), we display the PDF and CDF of reflectance ratio of 2.13 to 0.645 µm ($\frac{R_{2.13}}{R_{0.645}}$) to further emphasize the spectral features of different cloud phases (Figs. 10b1 and b2). As expected, the PDF of liquid-only clouds extends

to large reflectance ratio regions, and same as for ice-above-liquid cloud (Fig. 10b1). Moreover, there are about 60% of liquid-only cloud and 50% of ice-above-liquid cloud with reflectance ratio greater than 0.4 (Fig. 10b2), indicating that the reflectance ratio of ice-above-liquid cloud is in accord with low-level liquid cloud because of the thin features of the overlying ice clouds as discussed in Sect. 3.1.4 and 3.1.5. In contrast, ice-only clouds have very small frequencies when

reflectance ratio greater than 0.4 (Fig. 10b1). Mixed-only and ice-above-mixed clouds show even smaller reflectance ratio than ice-only clouds because more ice mass in the column results in stronger absorption at 2.13 µm and larger reflectance at 0.645 µm. For CC clear sky, the PDF shows its mode at the ratio ~ 0.1 with its width ranging from 0.0 to 0.4. Note that the mode of $R_{0.645}$ PDF locates at 0.02, and the mode of reflectance ratio PDF at 0.1 indicates that $R_{2.13}$ PDF

peaks closely at 0.002 which agrees with our examination and conforms the darkness of ocean surface at the SWIR. We also aware as the reflectance ratio of CC clear sky becomes larger, the corresponding $H_\sigma$ increases as well, indicating that the undetected liquid clouds in clear sky pixels (Sect. 3.1.5) also enlarge the SWIR to the visible reflectance ratio.

       At 1.375 µm, liquid-only clouds and clear sky show near zero average reflectance as the

photons are nearly all absorbed by water vapor. The mixed-only clouds have the largest average reflectance of 0.24 compared to other cloud groups, which is twice greater than that of ice-above-mixed clouds (~0.12). Much larger $R_{1.375}$ of mixed-only than ice-above-liquid clouds could be due to the reason that the former is much higher and thicker than the latter as shown in Fig. 4 and Table 4. Also, the average $R_{1.375}$ of ice-only clouds (~ 0.06) is greater than that of

ice-above-liquid clouds (~ 0.04). Because the specific humidity at the upper troposphere is nearly the same for both cloud groups (Fig. 5), larger $R_{1.375}$ of ice-only clouds demonstrates an average larger thickness than the ice layers above liquid clouds. This agrees with the results from the CC observations as discussed in Sect. 3.1.4 (Table 4).

       At the IR, the average $BT_{8.5}$ and $BT_{11}$ have similar magnitudes (Fig. 9). Because

atmospheric absorption at 8.5 and 11 µm is small and hence, the radiance reaching to the sensor above Ocean largely depends on the surface temperature in clear sky and cloud property in cloudy sky. Only the PDF and CDF of $BT_{11}$ are shown in Figs. 10c1 and c2, respectively. Note that when a cloud is opaque, the BT observed at the TOA only depends on cloud top, i.e. smaller BT for higher cloud top. As expected, the average $BT_{11}$ of mixed-only clouds is low (~ 231 K)

because these clouds are thick and high (Figs. 4, 6). Ice-above-mixed clouds have average $BT_{11}$ of ~ 247 K, which is larger than mixed-only clouds because ice-above-mixed clouds are optically thinner (as demonstrated by the reflectance at 0.645 µm) and have lower mixed layer tops (Table 4). The widths of $BT_{11}$ PDF of mixed-only and ice-above-mixed clouds are broad, but the frequencies are low when $BT_{11}$ > 260 K with about 20% samples at that $BT_{11}$ region (Fig. 10c2).

For liquid-only clouds, the average $BT_{11}$ (~ 288 K) is only slightly smaller than that of clear sky (~ 294 K) due to the low liquid cloud top (mode ~1 km, Fig. 6f) and thin features. The peak of $BT_{11}$ PDF of liquid-only clouds is consistent with that of clear sky ($BT_{11}$ ~ 294 K). For ice-above-liquid clouds, the average $BT_{11}$ (~ 274 K) is slightly larger than ice-only clouds (~ 272 K). This may due to that the ice layers above liquid clouds do not absorb as strong as ice-only clouds





because the former is averagely thinner than the latter over Ocean (Table 4). Also, the peaks of $BT_{11}$ PDF of ice-only and ice-above-liquid clouds are close to that of clear sky, and 50% samples of these two cloud groups with $BT_{11} > 280$ K (Fig. 10c2), demonstrating the thin features of these clouds which agree with the conclusions from the CC data (Fig. 6) and $R_{0.645}$ analysis.

5       Another notable feature of the BT shown in Fig. 9 is that the average $BT_{8.5}$ is averagely larger than the average $BT_{11}$ for ice-only, ice-above-liquid, ice-above-mixed and mixed-only clouds, i.e. 'ice-contained clouds', and vice versa for clear sky and liquid-only clouds. This is because in clear sky, absorption by the atmosphere at 8.5 μm is slightly greater than at 11 μm and hence, negative BTD between 8.5 and 11 μm ($BT_{8.5}$-$BT_{11}$) is observed (Fig 10d1). In cloudy
sky, cloud absorption at 11 μm is larger than at 8.5 μm (Wolters et al., 2008), which decreases the $BT_{11}$. The absorption difference between 11 and 8.5 μm is small for water, which explains that the PDF of liquid-only clouds locates at larger but still negative BTD regions (Fig. 10d1). That is, the PDF mode of liquid-only clouds (BTD ~ -1.8 K) is slightly larger than clear sky (BTD ~ -2.1 K), and the CDF of liquid-only clouds is very close to clear sky but slightly towards larger BTD regions (Fig. 10d2). Ice cloud has larger absorption difference between 11 and
8.5 μm than liquid cloud (Wolters et al., 2008). This explains positive BTD values for the 'ice-contained clouds'. For the BTD PDF of ice-above-mixed (peak at BTD ~1.2 K) and mixed-only clouds (peak at BTD ~ 1.5 K), there are only a small fraction of their samples (~20%) with BTD smaller than zero. Ice-only and ice-above-liquid clouds show very similar shape of BTD PDFs with their peaks located at BTD ~ -1.4 K. However, there are a lot of samples (> 50%) with BTD
greater than zero. The negative BTDs of ice-only or ice-above-liquid clouds also indicate their thin features over the SEA regions so that the absorption by ice clouds are insufficiently significant to produce positive BTD.

       For seasonal variations (Fig. 9), ice-only clouds have larger average $R_{0.645}$ in summer
due to that these clouds occur more frequently and thicker in this season. In contrast, liquid-only clouds over ocean have similar cloud tops but occur more frequently in winter and thus they show greater average $R_{0.645}$ in winter than in summer. For ice-above-mixed or mixed-only clouds, they have larger reflectance in summer agreeing with larger frequencies but smaller BT in fall consistent with higher cloud top (Table 4).

30       In summary, mixed-only and ice-above-mixed clouds are bright in the visible ( i.e. large $R_{0.645}$) and cold in the infrared (i.e. low BT). These clouds also have small reflectance ratio and positive BTD between 8.5 and 11 μm. Although liquid-only and ice-only clouds have similar $R_{0.546}$, liquid-only clouds tend to have relatively larger $BT_{11}$ and larger reflectance ratio than ice-only clouds. Ice-above-liquid clouds show slightly larger $R_{0.645}$ than either liquid-only or ice-
only cloud with reflectance ratio similar to liquid-only clouds, but $BT_{11}$ and BTD closer to ice-only clouds. The spectral radiative features of ice-only, liquid-only and ice-above-liquid clouds also demonstrate widespread thin clouds over SEA.

### 3.2. Cloud Phase Variations Associated with The Madden-Julian Oscillation

       Satellite-observed OLR has been widely used to characterize intraseasonal oscillations
either globally or regionally (e.g., Knutson et al., 1986; Wheeler and Hendon, 2004). The MJO active phase is well documented to associate with strong convection and precipitation that feature with small OLR (Zhang, 2005). Whereas in inactive phase, convective clouds and anvil cirrus are reduced as convection is suppressed and other cloud types will occur (Riley et al.,


2011; Zhang, 2005), leading to larger OLR (Zhang, 2005). Because ice, liquid clouds and their overlaps have different degree of inhomogeneity as shown in Sect. 3.1.5, cloud types varying with MJO phases not only alter radiation but will also modify the spatial heterogeneity observed at the TOA, which will be examined in this section.

5       As seasonality is a basic feature of the MJO (Zhang and Dong, 2004), we first classify the MJO events during 2007-2010 into four seasons according to the MJO index from Wheeler and Hendon (2004) (Fig. 11). In total, 917 events with amplitude greater than one are selected to represent strong MJO. As shown in Fig. 11a, the occurrence of different MJO phases displays very strong seasonality. Specifically, more MJO events happen in Phase 1 and 2 in summer,
while in fall, the MJO cases are concentrated in Phase 4, 5 and 6. When it moves to winter, the MJO prefers to occur in Phase 7, while in spring, more cases occur in Phase 1 and 8. Although the MJO amplitude is relatively flat across different phases, the weakest amplitude occurs in summer, while it is stronger towards winter, which is consistent with the statements in previous studies (Adames et al., 2016; Zhang and Dong, 2004).

To investigate the spatial heterogeneity associated with MJO, Figure 12 shows the spatial distributions of all-sky $H_\sigma$ over eight MJO phases. According to the spatial heterogeneity signatures derived in Sect. 3.1.5, the active MJO phase associated with deep convections and ice clouds is featured with small $H_\sigma$ values. Area with small $H_\sigma$ is surrounded by relatively large $H_\sigma$ values, indicating the locations where occur suppressed MJO phase associated with more liquid
clouds. As displayed by Fig. 12, the $H_\sigma$ pattern well reveals the propagation of MJO. That is, the center of convection associated with small $H_\sigma$ is in the Indian Ocean in Phase 1 and 2, while the West of Pacific and Maritime Continents are with large $H_\sigma$ as convection is suppressed. The small $H_\sigma$ pattern approaches to the Maritime Continent (indicated by the red dashed box in Fig. 12) in Phase 3 and centers at those regions in Phase 4, 5 and 6. After Phase 6, the connective
center with small $H_\sigma$ enters West Pacific Ocean, and at the same time, $H_\sigma$ becomes large over Indian Ocean. Note that Phase 2 mainly occurs in summer, which relates to boreal summer monsoon, so the $H_\sigma$ pattern here is quite similar with that in summer shown in Fig. 8g2. Similarly, the $H_\sigma$ pattern of Phase 5 is in concert with the seasonal pattern in fall (Fig. 8g3) due to a high chance of Phase 5 occurring in that season. The preference of geographical location of
MJO, featured by the small $H_\sigma$, moves from North of the equator in Phase 1 to the South of the Equator in Phase 6, indicating the seasonal cycle of MJO location.

To investigate more details of how $H_\sigma$ and cloud phase change with the MJO evolutions, Figure 12i shows the PDF of all-sky $H_\sigma$ sampling over the Maritime continent for the eight MJO phases from MODIS data. Two dashed lines from left to right indicate $H_\sigma \sim 0.01$ and $H_\sigma \sim 0.4$,
which are close to the mode position of $H_\sigma$ PDF of CC clear sky and liquid-only clouds (Fig. 7a), respectively. Figures 12j and k show the average occurrence frequency of different cloud phases from the MYD06 and the 2B-CLDCLASS-LIDAR products in the same region. In Phase 8 (cyan) when depressed MJO occurs over the Maritime continent, the $H_\sigma$ PDF shows the largest frequency ($\sim 0.38\%$) among all MJO phases at $H_\sigma \sim 0.4$ but the smallest frequency when $H_\sigma <$
0.01 (Fig. 12i). This corresponds to the largest frequency of clear sky and liquid clouds among all MJO phases but the smallest frequency of ice clouds indicated by MODIS (Fig. 12j) and of ice-only, ice-above-liquid, ice-above-mixed clouds displayed by the 2B-CLDCLASS-LIDAR products (Fig. 12k). In contrast, when active MJO phase locates in the Maritime continent (Phase 4, 5), the $H_\sigma$ PDF has the largest frequency at $H_\sigma < 0.01$ and the smallest frequency at $H_\sigma \sim 0.4$,





contributing by the frequent occurrence of ice or mixed clouds among all MJO phases but less
frequent occurrence of clear sky and liquid clouds.

Overall, the eastward-propagating $H_\sigma$ patterns-a behavior similar to OLR pattern (e.g.,
Wheeler and Hendon, 2004), indicate that the $H_\sigma$ could be useful for MJO studies such as
serving as an observed-based parameter that are sensitive to cloud phase, to track MJO position
and validate MJO simulations in climate models.

### 3.3. Interannual Variations: El Niño Southern Oscillation

The ENSO phenomenon has been known to dominate the interannual variability of
precipitation and clouds over the Western equatorial Pacific (As-Syakur et al., 2016; Reid et al.,
2012). In this section, the ENSO index shown in Fig. 14a is used to identify the warm (El Niño)
and cold (La Niña) phase of ENSO. Anomalies of clouds, radiation and heterogeneity in El Niño
and La Niña years are calculated based on the MODIS data from 2003-2017. The spatial
distributions of theses anomalies over SEA are shown in Fig. 13 and their time series are
displayed in Fig. 14. As expected, the spatial distributions show negative $R_{0.645}$ anomaly (red) in
El Niño year (Figs. 13a1-a4) and positive anomaly (blue) in La Niña year (Figs. 13b1-b4).
Opposite patterns are obtained for the $BT_{11}$ (Figs. 13c, d) and all-sky $H_\sigma$ anomaly (Figs. 13e, f).
The $H_\sigma$ anomaly indicates that more ice and mixed clouds (corresponding to positive $R_{0.645}$ and
negative $BT_{11}$ anomaly) occurring in La Niña year cause the spatial heterogeneity to be more
homogeneous than normal and vice versa in El Niño year. Also, the anomalies over the Maritime
continent tend to be stronger in winter and spring than in summer and fall.

Figure 14 displays the time series of monthly anomaly of clouds, radiances and spatial
heterogeneity for the areas indicated by the red-dashed box in Fig. 13a where are sensitive to
ENSO signals. As displayed, ice clouds detected by MODIS show similar variations as MODIS
all clouds with their anomalies ranging from -0.2-0.2, being positive in La Niña year and
negative in El Niño year (Fig. 14b). The MODIS ice cloud anomaly agrees with that of ice-only,
ice-above-liquid, ice-above-mixed and mixed-only clouds from the CC observations (Fig. 14d),
because most of these CC clouds are reported to be ice by MODIS (Table 2). Conversely, liquid
cloud occurrence is abnormally high (low) in El Niño (La Niña) year based on both the MODIS
and CC data (Figs. 14b, d). $H_\sigma$ is observed to be abnormally small in La Niña year due to the
increase of 'ice-contained clouds' and abnormally large in El Niño year because of the decreased
'ice-contained clouds' and the expose of liquid clouds (Figs. 14b,d). Moreover, the anomaly of
MODIS all and ice clouds ranges within 0.2 and their correlation is about 0.97 (significant at
99% confidence level), yet the magnitude of liquid cloud anomaly (within 0.06) seen from both
MODIS and CC is much smaller than that of ice clouds, indicating that the interannual variation
of clouds over SEA is dominated by the change of 'ice-contained clouds'.

Note that abnormally low (high) CC liquid-only or MODIS liquid clouds associated with
La Niña (El Niño) phase doesn't mean that total liquid clouds occur less (more) in La Niña (El
Niño) year because the CC data indicates the frequency of liquid clouds with ice clouds
overlying of 21.9% (Table 3). The overlying clouds can conceal liquid clouds to be observed
from space by passive sensors. Unlike CC liquid-only or MODIS liquid clouds, the CC ice-
above-liquid clouds occur abnormally high in La Niña year (Fig. 14d). When adding up the
frequency of liquid-only and ice-above-liquid clouds (i.e. total CC liquid cloud frequency), the
anomaly does not show clear relationship with ENSO (the subfigure in Fig. 14d). For example,





the La Niña phase in 2007 winter through 2008 spring generally accompanies abnormally high total CC liquid cloud occurrence, but in another La Niña year of 2008 winter through 2009 spring, the anomaly is generally negative. In Park and Leovy (2004), they showed negative anomalies of low-level clouds during positive ENSO phase using ship observations reported by the Extended Edited Cloud Report Archive (EECRA), i.e. less low-level clouds in El Niño year. While in our study, it is likely that MODIS data and four-year CC data is insufficient to support the relationship between liquid cloud occurrnece and ENSO over SEA.

Overall, interannual variations of clouds over SEA are primarily due to the change of ice clouds, which are consistent with the statements in previous studies that in El Niño year, anomalous subsidence over the Western equatorial Pacific decreases cloud amount by suppressing deep convection and reducing high clouds (Park and Leovy, 2004; Wang and Su, 2015). As a result, the observed radiation at the TOA adjust accordingly (Loeb et al., 2012). As cloud phases vary interannually and hence change the spatial heterogeneity, i.e., being smoother in La Niña year than normal and vice versa in El Niño year. Also, the timeseries of all-sky $H_\sigma$ anomaly well correlates with that of $R_{0.645}$ (r ~ -0.8, significant at 99% confidence level) and $BT_{11}$ (r ~ 0.8, significant at 99% confidence level), indicating that $H_\sigma$ can be one valuable observed parameter to investigate ENSO such as tracking ENSO or validation of ENSO simulations by climate models.

## 4. Discussions and Conclusions

This work is in a series of studies that examine cloud vertical structures. Li et al. (2015) explored the vertical distributions of cloud types using the 2B-CLDCLASS-LIDAR data, while Oreopoulos et al. (2017) interpreted the overlap feature of high, middle and low clouds. Considering the needed cloud phase information for improving GCMs (e.g., Cesana and Storelvmo, 2017), the current study focuses on investigating the characteristics of cloud vertical structures, spatial heterogeneity and spectral radiances from the perspective of cloud phases over Southeast Asia. Utilizing the state-of-the-art CloudSat and CALIPSO (CC) observations, five cloud groups have been classified including ice-only, ice-above-liquid, ice-above-mixed, liquid-only and mixed-only clouds to capture the main vertical structures of cloud phases. By collocating the CC-MODIS data, the spectral and spatial heterogeneity signatures of each CC cloud groups have been examined. Seasonal, intraseasonal and interannual variations of these cloud phase characteristics have also been shown in this work.

A general review on cloud spatial distributions and meteorology shows that the annual cloud occurrence frequency over SEA is about 80.9% being more frequent in summer (85.8%) and less frequent in winter (77.7%) based on the CC observations. Ice-only (27.8%), ice-above-liquid (21.9%), ice-above-mixed (7.4%) and mixed-only (6.2%) clouds, i.e. 'ice contained clouds', prefer a warm, humid and unstable environment as these clouds occur associated with the seasonal movement of the monsoon and ITCZ climate systems. It is noted that ice-only and ice-above-liquid clouds occur associated with similar upper troposphere dynamics, i.e., comparable mean temperature, specific humidity and static stability, while ice-above-mixed clouds occur in an environment with larger specific humidity at the upper troposphere. Liquid-only clouds occur frequently in winter and spring over Southeast China and East China sea where are relatively cold, dry and stable.





It is shown that ice clouds over SEA are thin with more than 60% and 80% samples with geometrical thickness smaller than 3.0 km and optical depth less than 3. Ice-only clouds have larger mean thickness (geometrical and optical) and Re than the ice layers above liquid or mixed clouds. Although there could exist sampling biases due to instrument limitations, yet higher

frequency of ice-only clouds at $1.6 < \tau < 3.0$ demonstrates that more ice-only clouds develop thicker than either the ice layers above liquid or mixed clouds. The tops of liquid-only clouds are on average lower than that of the liquid below ice clouds. As liquid-only clouds occur more frequently with increase of LTSS, their vertical development is likely to be inhibited. However, ice-above-liquid clouds more favorably distribute in a warm, humid and unstable environment,

which allows the underlying liquid clouds to grow deeper. The tops of mixed layers below ice clouds are primarily located at about 6 km. For mixed-only clouds, their tops are primarily at 16 km over ocean, but over land, a large fraction of samples have cloud top around 6 km. These results indicate that ice-above-mixed and mixed-only clouds over land are under development in the early afternoon.

We also show that distinct spatial heterogeneity exists between different cloud phases and their overlaps. Ice-only, ice-above-mixed and mixed-only clouds are usually homogeneous, i.e., small $H_\sigma$ values. In contrast, liquid-only clouds show the largest $H_\sigma$ among all cloud phase groups, being the most heterogenous. Ice-above-liquid clouds have large but slightly smaller $H_\sigma$ values than liquid-only clouds because the overlying homogenous and thin ice clouds slightly

smooth the radiation emerging from the low-level liquid clouds. A typical $H_\sigma$ value for clear sky is 0.01, however, clear sky $H_\sigma$ can be as large as 1, resulting from the increase of subpixel variability due to undetected liquid clouds in some MODIS pixels. As large $H_\sigma$ values are reported to associate with the biases of cloud $\tau$ and Re derived from passive sensors (Di Girolamo et al., 2010; Zhang and Platnick, 2011), these biases resulted from plane-parallel

assumption are larger for liquid than ice clouds. One should be cautious to these biases when interpreting cloud-aerosol relationship. The seasonal patterns of all-sky $H_\sigma$ show that small values are in north of the equator over Indian and West Pacific Ocean in summer but move southward in autumn and winter, being consistent with the seasonal shift of the 'ice-contained clouds'.

Difference of cloud optical, micro and macrophysical properties leads to distinct spectral features between the five cloud groups. The liquid-only clouds show zero $R_{1.375}$, large reflectance ratio between the SWIR and the visible, high BTs and negative BTD between 8.5 and 11 μm. Ice-only clouds in contrast show notable $R_{1.375}$, reflectance ratio generally smaller than 0.4, relatively low BTs and more than 50% samples with positive BTD. Ice-above-liquid clouds

behave more like ice-only clouds in the IR because these two cloud groups have similar $BT_{11}$ and BTD PDFs, but ice-above-liquid clouds act more like liquid-only clouds in the SW spectral due to their similar PDFs/CDFs of reflectance ratio. The average $R_{1.375}$ of ice-above-liquid clouds, which is mainly contributed by the ice layers only, is smaller than that of ice-only clouds. Similarly, the average $BT_{11}$ of ice-above-liquid clouds is slightly larger than that of ice-only

clouds. These results demonstrate that ice-only clouds are on average thicker than the ice layers above liquid clouds, being consistent with the conclusions derived from the CC observations. Mixed-only or ice-above-mixed clouds usually have large $R_{0.64}$, small reflectance ratio, low $BT_{11}$ and positive BTD. It is also noted that the $R_{0.645}$ and $BT_{11}$ PDFs of ice-only, liquid-only





and ice-above-liquid clouds show their frequency peak nearly same as that of clear sky, revealing ubiquitous thin clouds over SEA.

Cloud phases together with their spectral and spatial heterogeneity features have also been examined in different MJO and ENSO phases. In the MJO active phase, more frequent 'ice-contained clouds' attribute to a smooth MJO center. On both sides of the MJO convective center, large $H_\sigma$ values are observed due to increase of liquid clouds and decrease of 'ice-contained clouds'. Similarly, the interannual variation of clouds over SEA is primarily due to the change of 'ice-contained clouds'. Increased 'ice-contained clouds' in La Niña year result in abnormally homogeneous spatial heterogeneity, stronger $R_{0.645}$ and lower $BT_{11}$, and vice versa in El Niño year. The observed $H_\sigma$ values capture the MJO and ENSO features, implying that the $H_\sigma$ is able to track MJO and ENSO and provides a way to validate their simulations in GCMs.

As the CC observations are sensitive to a wide range of cloud scenarios, cloud patterns shown here can help interpreting the cloud product variability in SEA as discussed in Reid et al., (2013). Climatological characteristics of cloud phases provided in this study can serve as a benchmark to improve the performance of climate models in validating their simulations of cloud phases and their vertical overlap, their spatial heterogeneity and spectral signatures. Particularly, spatial heterogeneity, a direct measured variable from satellite that reveals subpixel variability of different cloud phases, is not only able to track the MJO and ENSO but is also useful to evaluate how well GCMs capture subpixel clouds (e.g., Loveridge and Davies, 2019).

*Data availability.* Data used in this study are summarized in Table 1 and their availability is provided in acknowledgements.

*Author contributions.* YH and LD conceived this study. YH analyzed the results and wrote the manuscript. LD joined result discussions and refined this manuscript.

*Competing interests.* The authors declare that they have no conflict of interest.

*Acknowledgements.* This research has been supported by the Cloud, Aerosol and Monsoon Processes Philippines Experiment (CAMP[2]Ex) and by NASA Grant 80NSSC18K0144. We are grateful to Guangyu Zhao for his offer of clear sky $H_\sigma$ derived from ASTER and MODIS. We acknowledge the members of the CloudSat Data Processing Center who provide CloudSat products, including 2B-CLDCLASS-LIDAR, 2C-ICE and ECMWF-AUX, available at http://www.cloudsat.cira.colostate.edu/. The MODIS products, MYD021KM, MYD03 and MYD06, are obtained via the Atmosphere Archive and Distribution System (LAADS) Distributed Active Archive Center (DAAC), which is available at https://ladsweb.modaps.eosdis.nasa.gov/. We also thank Bureau of Meteorology in Australia offering MJO index and NOAA Earth System Research Laboratory providing Multivariate ENSO index.



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



**Tables**

Table 1. Summary of datasets used in this study.

| Products | Version | Parameters | Resolution (km) | Period | References |
|---|---|---|---|---|---|
| 2B-CLDCLASS-LIDAR | P_R04 | Cloud layers, phases | 1.4 x 1.8 | 2007-2010 | (Wang et al., 2012) |
| 2C-ICE | P1_R04 | Extinction coefficient, effective radius | 1.4 x 1.8 x 0.24 | 2007-2010 | (Deng et al., 2010) |
| ECMWF-AUX | P_R05 | Meteorology | 1.4 x 1.8 x 0.24 | 2007-2010 | (Cronk and Partain, 2017) |
| MYD021KM | C6.1 L1B | Reflectance, BT | 1 | 2003-2017 | (Savtchenko et al., 2004) |
| MYD03 | C6.1 L2 | Geolocation, Ocean/land Mask | 1 | 2003-2017 | (Savtchenko et al., 2004) |
| MYD06 | C6.1 L2 | Cloud Phase, spatial heterogeneity | 1 | 2003-2017 | (Savtchenko et al., 2004) |
| -- | | MJO index | -- | 2007-2010 | (Wheeler and Hendon, 2004) |
| -- | | ENSO index | -- | 2003-2017 | (Wolter and Timlin, 1998) |

Table 2. Comparison of MODIS and CC cloud phase. Number in parentheses represents the percentage (%) of CC phase reported as to be MODIS phase.

| MODIS cloud phase | CloudSat-CALIPSO cloud phase | | | | | | |
|---|---|---|---|---|---|---|---|
| | Clear | Ice only | Liquid only | Mixed only | Ice above liquid | Ice above mixed | sum |
| Clear | 2 193 923 (91.2) | 1 059 415 (29.2) | 597 838 (28.4) | 11 903 (1.40) | 305 350 (10.5) | 5 813 (0.6) | 4 174 242 |
| Liquid | 185 470 (7.71) | 227 625 (6.28) | 1 380 378 (65.6) | 67 412 (7.91) | 783 585 (27.0) | 40 179 (4.1) | 2 684 649 |
| ice | 23 576 (0.98) | 2 259 465 (62.31) | 87 086 (4.14) | 603 590 (70.9) | 1 636 624 (56.4) | 720 588 (74.3) | 5 330 929 |
| undetermined | 2 741 (0.11) | 79 482 (2.19) | 38 450 (1.83) | 168 921 (19.8) | 177 710 (6.12) | 202 513 (20.9) | 669 817 |
| sum | 2 405 710 | 3 625 987 | 2 103 752 | 851 826 | 2 903 269 | 969 093 | 12 859 637 |



Table 3. Cloud occurrence frequency (%) derived from 2B-CLDCLASS-LIDAR. The number outside parentheses represents both land and ocean. First number in parentheses is for ocean and the second one is for land.

|  | Ice only | Liquid only | Mixed only | Ice above liquid | Ice above mixed | Others | All clouds |
|---|---|---|---|---|---|---|---|
| MAM | 28.8(32.0,19.8) | 17.4(15.5,22.0) | 5.5(4.6,7.3) | 20.7(20.2,23.6) | 6.2(5.9,7.0) | 0.71(0.58,1.1) | 79.3(78.8,80.7) |
| JJA | 30.5(34.7,19.6) | 12.8(10.9,17.4) | 7.3(6.9,7.9) | 25.0(22.1,35.3) | 9.5(8.9,11.4) | 0.66(0.53,1.0) | 85.8(84.1,92.7) |
| SON | 28.1(32.9,17.8) | 15.9(12.4,22.8) | 6.6(6.1,6.9) | 21.7(22.5,23.0) | 7.8(8.3,7.7) | 0.70(0.54,1.0) | 80.8(82.8,79.2) |
| DJF | 23.8(26.5,16.4) | 21.5(20.5,22.9) | 5.4(5.2,5.0) | 20.3(21.8,19.1) | 6.0(6.4,5.7) | 0.74(0.72,0.69) | 77.7(81.2,69.7) |
| Annual | 27.8(31.3,18.4) | 16.9(14.8,21.3) | 6.2(5.7,6.8) | 21.9(21.7,25.3) | 7.4(7.4,8.0) | 0.70(0.60,0.94) | 80.9(81.7,80.6) |

5   Table 4. Average of cloud properties for the five cloud groups over ocean and land.

|  | Ice only | | | | | Liquid only | Ice above liquid | | | | | | Ice above mixed | | | | | | Mixed only |
|---|---|---|---|---|---|---|---|---|---|---|---|---|---|---|---|---|---|---|---|
|  | $T_i^1$ | $B_i^2$ | $H_i^3$ | $\tau_i$ | $Re_i$ | $T_w$ | $T_i$ | $B_i$ | $H_i$ | $\tau_i$ | $Re_i$ | $T_w$ | $T_i$ | $B_i$ | $H_i$ | $\tau_i$ | $Re_i$ | $T_m$ | $T_m$ |
| **Ocean** | | | | | | | | | | | | | | | | | | | |
| MAM | 14.9 | 11.5 | 2.9 | 2.4 | 31.3 | 2.0 | 14.8 | 11.8 | 2.4 | 1.8 | 28.5 | 2.9 | 15.0 | 12.0 | 2.5 | 1.9 | 27.9 | 9.4 | 13.1 |
| JJA | 14.9 | 10.9 | 3.5 | 3.0 | 32.6 | 2.2 | 14.9 | 11.5 | 2.7 | 2.1 | 29.2 | 3.2 | 15.2 | 11.9 | 2.7 | 1.9 | 27.4 | 9.4 | 14.2 |
| SON | 15.1 | 11.1 | 3.4 | 2.8 | 31.6 | 2.1 | 15.1 | 11.6 | 2.8 | 2.1 | 28.6 | 3.2 | 15.3 | 12.0 | 2.8 | 1.9 | 27.3 | 9.5 | 14.1 |
| DJF | 15.3 | 11.4 | 3.4 | 2.4 | 29.7 | 2.2 | 15.2 | 11.9 | 2.7 | 1.7 | 27.1 | 3.1 | 15.4 | 11.9 | 2.9 | 1.8 | 26.6 | 9.3 | 13.2 |
| **Land** | | | | | | | | | | | | | | | | | | | |
| MAM | 13.7 | 10.1 | 3.1 | 4.3 | 36.8 | 3.4 | 14.3 | 11.4 | 2.2 | 1.8 | 29.5 | 3.6 | 14.3 | 11.7 | 2.1 | 1.7 | 28.9 | 8.8 | 10.3 |
| JJA | 14.9 | 10.4 | 3.9 | 4.7 | 35.8 | 3.6 | 15.0 | 11.9 | 2.5 | 1.6 | 28.0 | 3.9 | 15.1 | 12.3 | 2.2 | 1.4 | 27.1 | 9.3 | 12.3 |
| SON | 14.5 | 10.6 | 3.4 | 3.8 | 34.1 | 3.4 | 14.8 | 11.7 | 2.5 | 1.9 | 28.3 | 3.7 | 15.0 | 12.0 | 2.4 | 1.8 | 27.8 | 9.2 | 11.1 |
| DJF | 13.7 | 10.0 | 3.2 | 3.5 | 33.5 | 2.9 | 14.8 | 11.5 | 2.6 | 1.5 | 26.9 | 3.6 | 15.0 | 11.8 | 2.6 | 1.5 | 26.0 | 8.9 | 9.3 |

1: T for cloud top
2: B for cloud base
3: H for geometric thickness
Subscripts: i for ice, w for water and m for mixed





## Figures

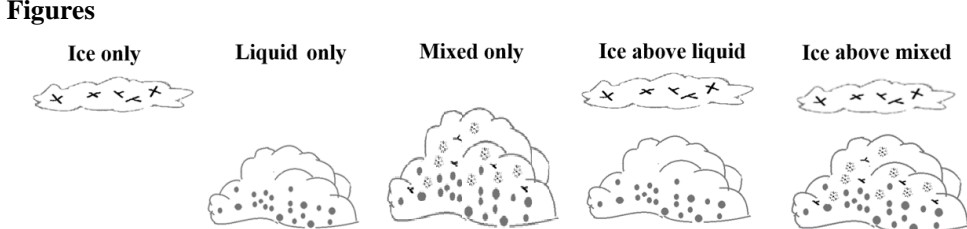

Figure 1. Schematic of cloud classification based on the 2B-CLDCLASS-LIDAR product.

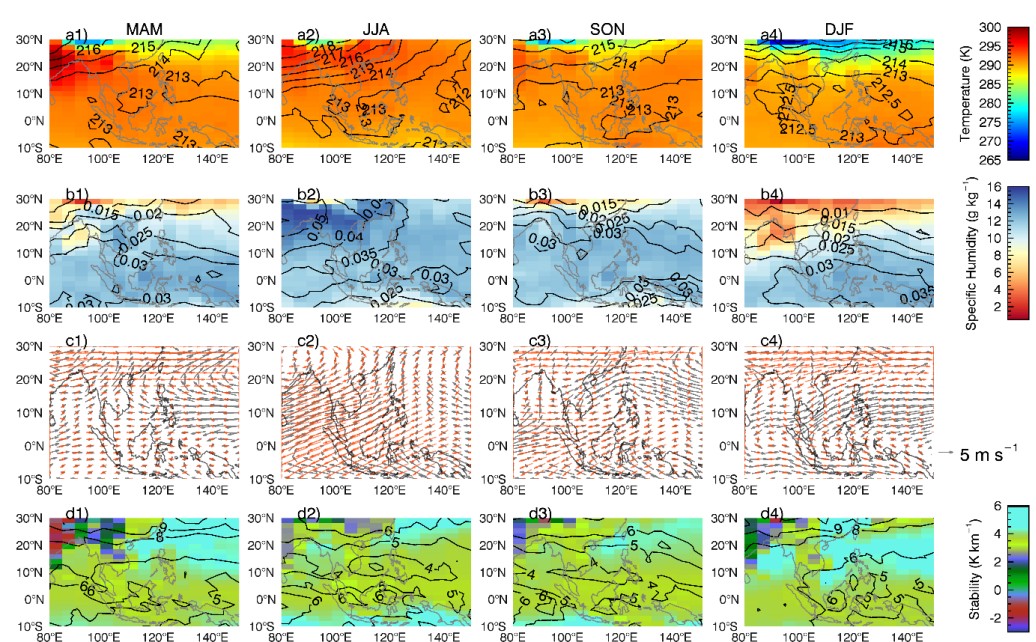

Figure 2. Temperature (a1-a4), specific humidity (b1-b4), wind field (c1-c4), and static stability (d1-d4) derived from the ECMWF-AUX data: shade for 850 hPa, contour for 180 hPa, grey vectors for wind field at 850 hPa and red vectors for 180 hPa.


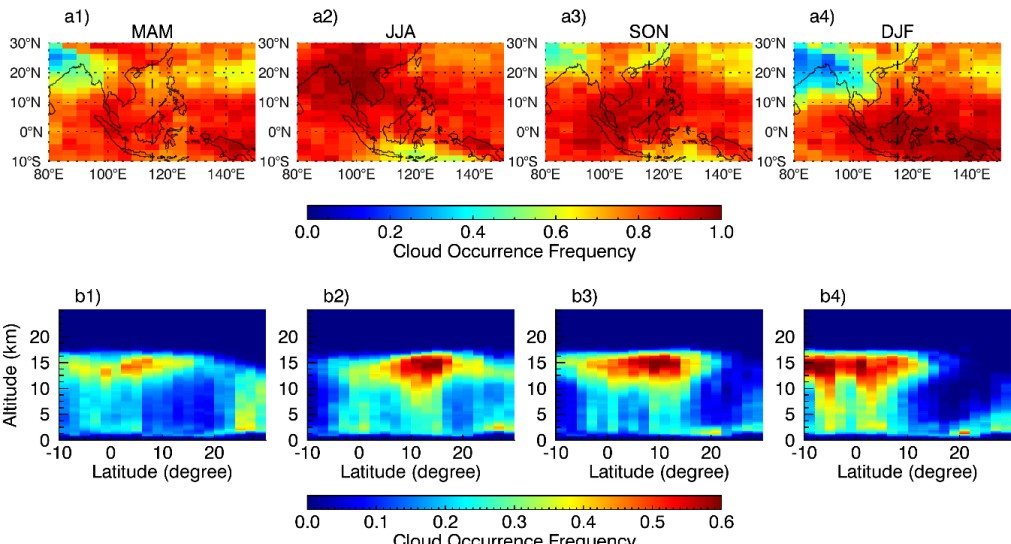

Figure 3. Cloud occurrence frequency derived from 2B-CLDCLASS-LIDAR: a1) -a4) for horizonal distribution and b1-b4) for latitude-altitude cross section at the longitude of 115°E, indicated by the dashed line at the upper panels.





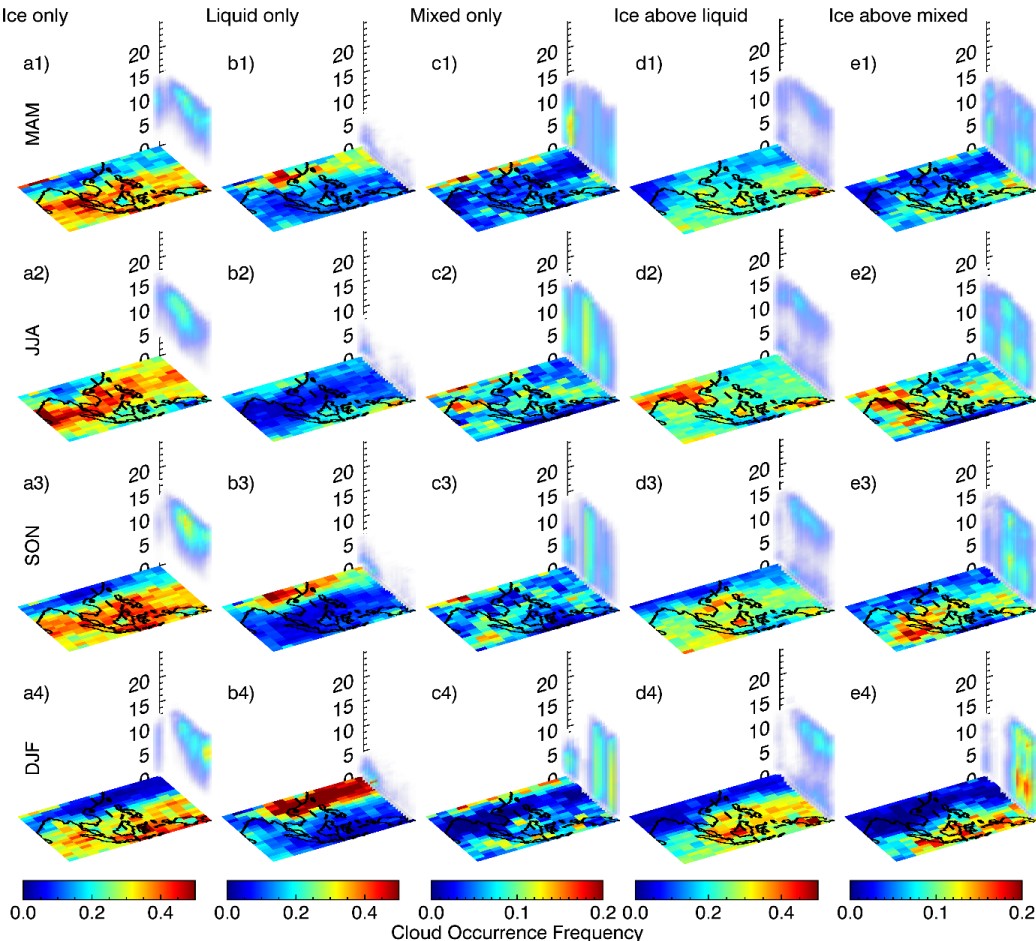

Figure 4. Occurrence frequency of the five cloud groups derived from 2B-CLDCLASS-LIDAR. Note that colorbar scale is different for the five cloud groups. Cross-section is for the longitude centered at 115°E indicated by the black dashed line. Zero values of vertical frequency are set to be transparent for a better view.





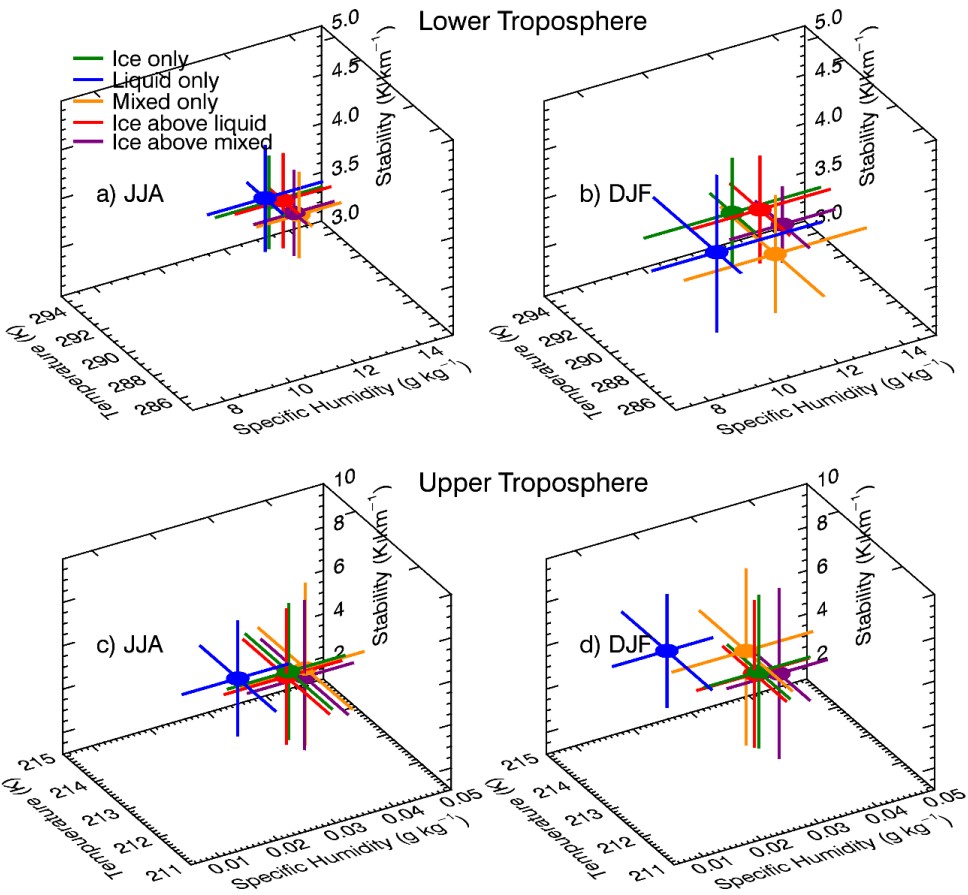

Figure 5. Mean and standard deviation of meteorological variables over ocean for the five cloud groups.





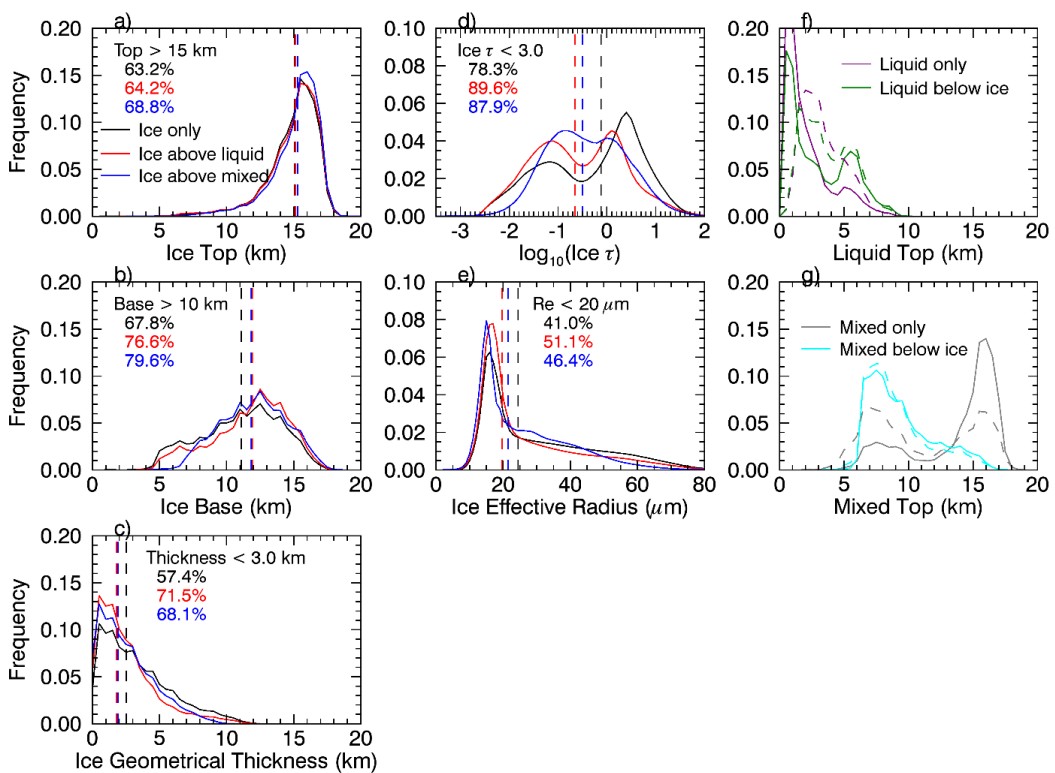

Figure 6. Annual PDFs of a) cloud top, b) base, c) geometric thickness, d) optical depth and e) effective radius for ice clouds including both ocean and land; f) cloud top for liquid-only cloud and the liquid below ice and g) cloud top for mixed-only cloud and the mixed below ice. In f) and g), solid and dashed lines are for ocean and land, respectively. Cloud top and base bins adopt an interval of 0.5 km. Ice τ and Re bins use an interval of 0.1 in logscale and 1 μm, respectively.



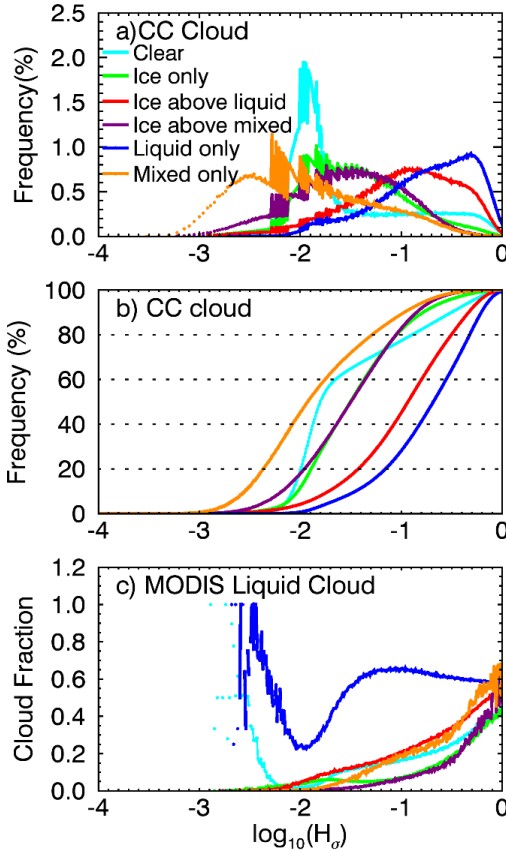

Figure 7. a) Annual PDFs of spatial heterogeneity for CC clear sky and the five cloud groups; b) same as a) but for CDF; c) liquid cloud fraction in the 5x5 surrounding of the collocated CC-MODIS pixel derived from the MYD06 IR cloud phase retrievals. $H_\sigma$ bin interval is 0.01 in log scale.

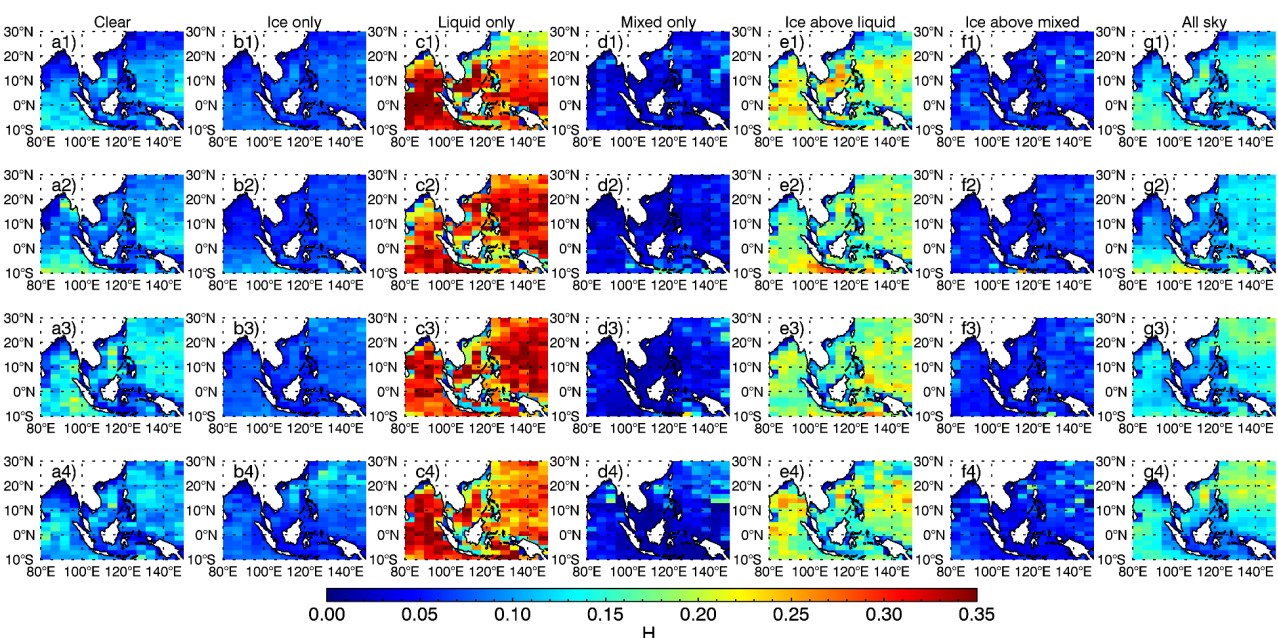

Figure 8. Spatial distributions of $H_\sigma$ for clear sky, the five cloud groups and all sky: from top to bottom panels for MAM, JJA, SON and DJF, respectively.



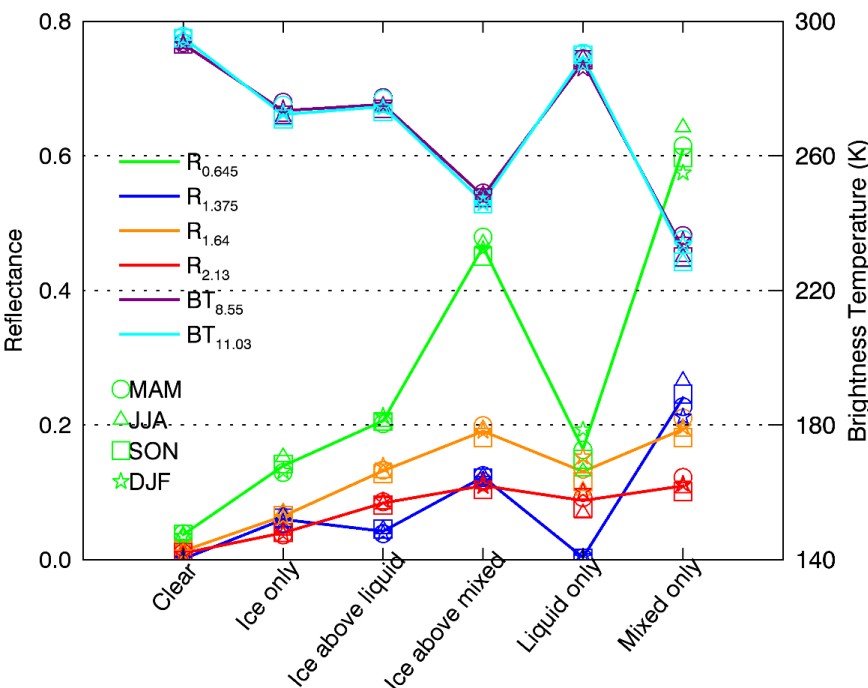

Figure 9. Average reflectance and brightness temperatures over ocean for clear sky and the five cloud groups, including all solar zenith angles. The sold lines represent annual mean and symbols denote seasonal averages.

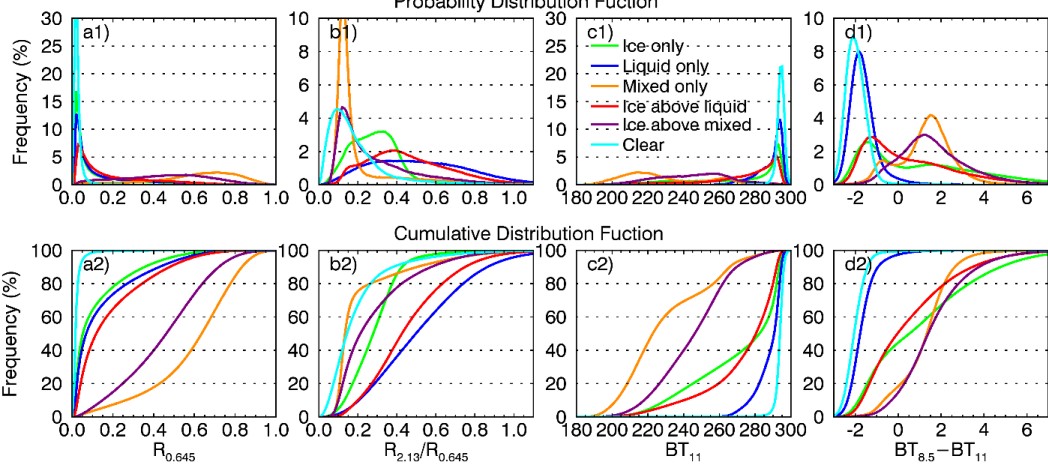





Figure 10. The annual PDFs and CDFs of reflectance at 0.645 μm, reflectance ratio, BT at 11 μm and BTD between 8.5 and 11 μm. The intervals for reflectance, reflectance ratio, BT and BTD are 0.01, 0.01, 1 K and 0.1 K, respectively.

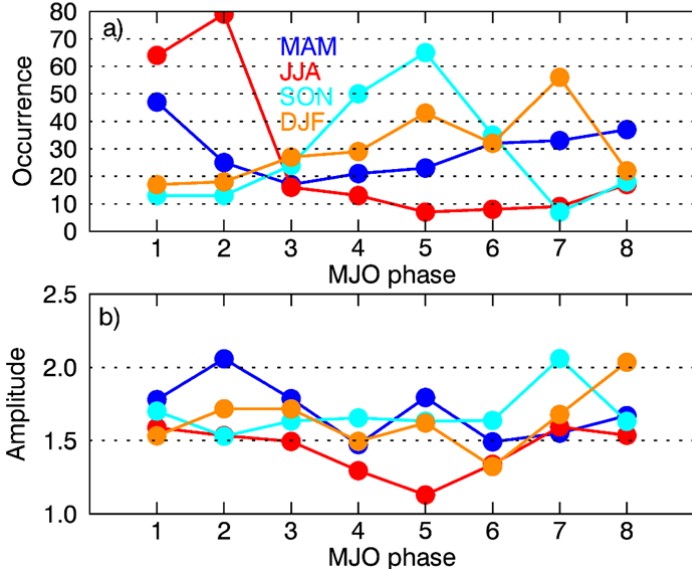

5     Figure 11. A statistical summary of strong MJO phases with amplitude > 1 from 2007-2010.

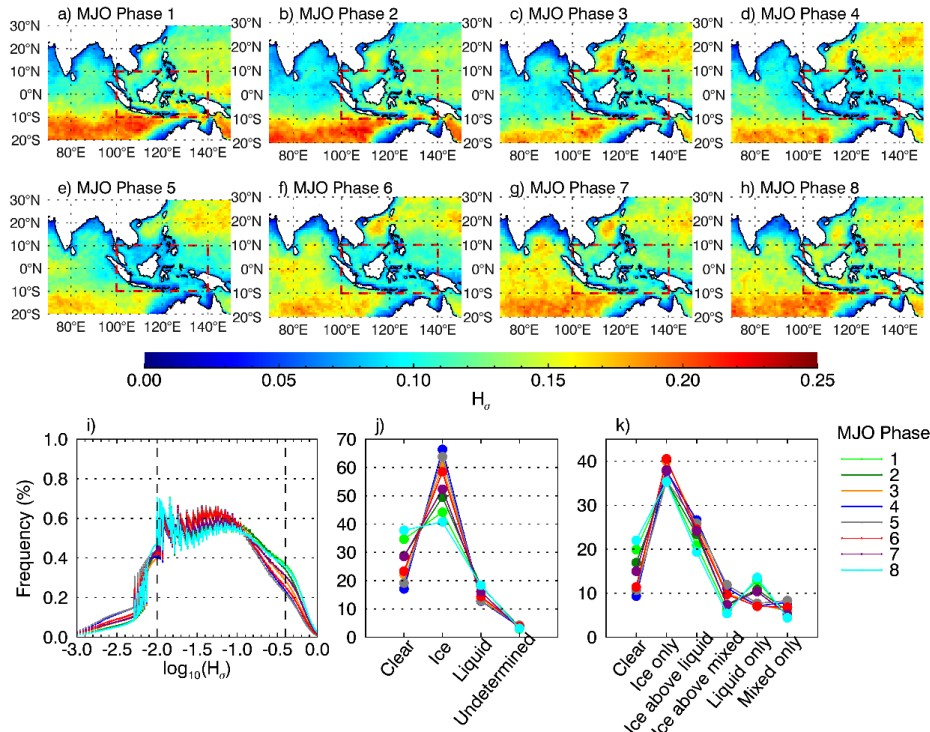

Figure 12. Cloud phase characteristics in different MJO Phases: a)-h) spatial distributions of all-sky $H_\sigma$ derived from MYD06; i) PDF of all-sky $H_\sigma$ derived from MYD06 over the Maritime continents (the red dashed box in each panel); j) occurrence frequency of clear sky, ice, liquid and undetermined clouds from MYD06 (unit:%); k) same as j) but for clear sky and five groups derived from the 2B-CLDCLASS-LIDAR product.

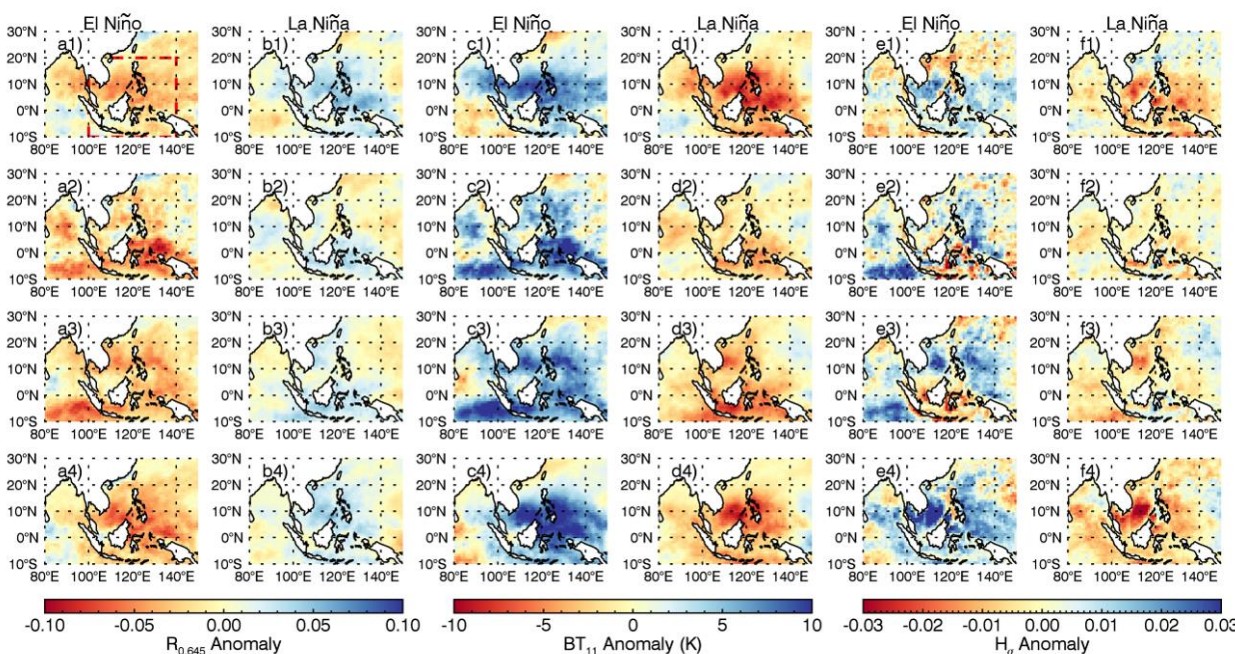

Figure 13. Reflectance, brightness temperature and spatial heterogeneity anomaly in El Niño and La Niña year: from top to bottom for MAM, JJA, SON and DJF.



Figure 14. ENSO index and monthly anomaly of cloud, spatial heterogeneity and radiation in the region represented by the dashed-red box in Fig. 13a1.

