# Peer review of "Cloud Phase Characteristics Over Southeast Asia from A-Train Satellite Observations"

_Atmospheric Chemistry and Physics, 2019_

## Referee Comment (RC1) · Anonymous Referee #2 · 24 Jan 2020

General comment

This study examines characteristics of cloud phases in five, frequently occurring, overlapping configurations, over a wide area of southeast Asia. For this purpose, the authors use combined CloudSat-CALIPSO and MODIS data. The different cloud phases are examined in terms of their seasonality and relationship with meteorology, and frequency of occurrence. Their heterogeneity and spectral radiance characteristics are examined in combination with corresponding MODIS data. Associations with MJO and ENSO are also investigated.

The study is to a large extent comprehensive. The results are discussed adequately,

and the findings combine verification of previously known characteristics of specific cloud phases/types and their combinations, with new insights over their future usefulness in field campaigns and GCM evaluations. For these reasons, I recommend acceptance of this manuscript for publication in ACP. I include a list of minor comments and technical corrections for the authors to consider.

Comments

Page 5, lines 25-28: it would be useful to report how often these "multi-layer, same phase" vertical structures occur, and discuss possible consequences of this simplification.

Page 6, lines 24-25: I don't understand how MODIS detects less clear-sky cases than CC by missing some cloudy cases. Shouldn't it be the other way around?

Figure 5: this figure is hard to read. Please consider replacing with 2D plots.

Page 16, lines 26-27: how is the frequency of occurrence related to the average reflectance? Shouldn't they be thicker to have higher R?

Page 18, lines 25-27: it is hard to verify this statement based on Fig. 14d. For example, ice-above-liquid after 07/08 does not agree well.

Page 18, lines 29, 30: what is considered "abnormal" in the heterogeneity index variation?

Page 19, line 41: "where are relatively cold". Are you referring to the lower troposphere conditions? Please clarify.

Figure 6: what are the vertical dashed lines?

Figure 9: seasonality symbols are not clear. Please consider plotting differently or including a table.

Corrections

Page 2, line 14: "macrophyscial" should read "macrophysical".

Page 2, lines 28-31: please consider rephrasing or breaking this long sentence.

Page 3, lines 8-9: do you mean "has not yet been examined"?

Page 9, line 1: "cloud" should read "could".

Page 10, line 4: "summaries" should read "summarizes".

Page 11, line 22: "CLIPASO" should read "CALIPSO".

Page 13, line 3: please omit "that".

Page 13, line 42: "spatial" should read "spatially".

Page 14, line 2: "it" should read "its".

Page 14, line 35: "reflected" should read "reflective".

Page 14, lines 42-43: do you mean "refractive index"?

Page 15, line 16: please consider replacing "aware" with e.g. "note that".

Page 16, line 25: "and thicker" should read "and they are thicker".

Page 16, line 33: 0.546 should read 0.645.

Page 17, lines 18-20: please rephrase.

Page 17, line 22: please consider replacing "are with" with e.g. "display".

Page 17, line 24: "connective" should read "convective".

Page 19, lines 13-14: please rephrase.

Page 19, line 15: "well correlates" should read "correlates well".

Page 20, line 18: "heterogenous" should read "heterogeneous".

Page 21, line 5: please replace "attribute" with "contribute".

---

## Referee Comment (RC2) · Anonymous Referee #3 · 28 Jan 2020

In this paper, the authors investigate cloud properties as seen by different A-train satellites over the Southeast Asia. They further divide clouds into 5 cloud types as a function of their cloud phase and different overlapping possibilities among the cloud-type layers. In the last part of the manuscript, they study possible links between these 5 types and MJO and ENSO conditions.

While the topic of this paper aligns with the scope of the journal, it is difficult to judge the novelty of the analysis since most of it is a kind of climatology rather than new results. However, I acknowledge a tremendous amount of work from the

authors. Yet, the paper is too long and descriptive, which makes it difficult to follow. In addition to trimming the manuscript, I have a couple more concerns to address before recommending this paper for publication. A more detailed explanation is provided below.

Main concerns

1) This paper is too long and descriptive. It is hard to follow and I often lost track of the goal of the sections. Every section should be reduced in size and I would recommend focusing on specific findings relevant to the topic of the study rather than describing every subplot of the figures as well as the behavior of each cloud types...

2) If I understand correctly there is no filtering of the data whatsoever to make them consistent with each other. I find this a little concerning. For example, in section 3.5.1 when comparing the spatial heterogeneity index with CloudSat-CALIPSO, all pixels are used including those where CC and MODIS cloud masks disagree. This may result in large biases as explained by the authors later on. It would be best to keep in the main analysis the pixels where CC and MODIS agree.

3) The authors use the version R04 of the 2B-CLDCLASS-LIDAR product. This version is not free of uncertainties in particular when it comes to detecting shallow cumulus clouds. It's been shown that this version overestimates the amount of shallow cumulus clouds (https://www.earth-syst-sci-data.net/11/1745/2019/essd-11-1745-2019.html). Similarly, nothing is said about any kind of uncertainty in the cloud phase retrieval of this product. For example, has this product been evaluated against other cloud phase dataset (ground-based, satellite or in situ?). I know that the cloud phase confidence considerably decreases when the lidar is totally attenuated and the decision tree only relies on the radar signal. I would suggest the authors to mention these at least.

4) Finally, the authors consistently mention that their results could be used for model evaluation but fail to explain how. I understand it is tempting to sell any observational result as a possible constraint for model, but if the authors want to do so, they need to explain how and why, which is not done here.

Minor comments
Throughout the manuscript, the authors use plural with the term cloud phase, I feel like most of the time, it would be better to use singular.

P2 L35: improve GCM performance $\Rightarrow$ improve climate simulations
P4 L8: in the lower troposphere $\Rightarrow$ below 8 km
P4 L8: in the upper troposphere $\Rightarrow$ above 8 km
P4 L21: Please specify the version. From your table, I believe you use the version R04.
P4 second paragraph: You don't describe how the algorithm works when the lidar signal is completely attenuated. The cloud phase is then based only on Ze and T thresholds, which substantially decreases its confidence level. Since the region under study is dominated by convective clouds, this situation may occur very often.
P8 L13: I don't understand the unit of LTS, it's supposed to be in K (or ËŽC).
P8 L18-23: The CLDCLASS-LIDAR product provides a cloud fraction (between 0 and 1) per layer so how do you get cloud and sample numbers?
P8 L25: Large $\Rightarrow$ large
Fig. 3: why do you use this particular cross section rather than a zonal mean.
P8 L41: "Also... of cirrus". Why do you mention MISR here out of the blue? Also this sentence is confusing.

P9 L1: Can you elaborate on this statement? DO you mean for that region?

P9 L19: Confusing sentence.

P11 L13: This should appear in the data section along with the other uncertainties related to the datasets.

P11 L22: CLIPASO $\Rightarrow$ CALIPSO

P11 L22: attenuated by clouds with optical thickness greater than 3.

P12 L14: spatial $\Rightarrow$ spatially

P12 L26: "due to the small optical thickness of the.."

P12 L27: It's unclear to me why the authors constantly refer to MISR for no reasons since MISR observations are used in this study.

P12 L36: This sentence needs re-wording.

P13 L1-2: I would strongly recommend excluding pixels in which CC and MODIS disagree in the main figures rather than only mentioning it as "not shown".

P13 L2-4: There is no main verb in this sentence, please re-word. Also, are you referring to shallow cumulus clouds? In this case it would be rather easy to validate your hypothesis by focusing on a shallow Cu dominated region, such as the Barbados. However, depending on the MODIS product used, Pincus et al 2012 reported that a substantial amount of these clouds (partially-filled pixels) are excluded of the cloud product. Another important thing to note is that, R04 over-estimate shallow Cu cloud fraction (see main concern comments).

P13 L33: Proves seems a bit strong.

P14 L35: reflecting

P17: Why are you showing MJO phases? It's been documented in many many studies already. What does this bring to your study?

P17 L38: "suppressed"

P18 L3: I don't understand the meaning of this sentence and I don't see how this could be used to validate climate models.

P18 L8-20: It basically shows there are more convective clouds.

P19 L13-14: Another sentence without meaning, please re-word.

P19 L36: "preferentially occur in"

P19 L38: occur ⇒ are

P21 L10: a comma is missing after ENSO L11. Overall, I would suggest rephrasing this sentence because I don't think the authors can claim the heterogeneity index captures MJO or ENSO. At best, it varies for the different ENSO/MJO phases, but it's definitely not well correlated. For the second part of the sentence, unless the authors explain how one could use this for model evaluation, I'd recommend to remove.

P21 L14-16: Here again, there is no tool to compare this to models, at least to the best of my knowledge, so unless the authors elaborate on this statement, they should remove this statement. I can envision a qualitative comparison of heterogeneity at best.

---

## Author Response (AR1)

*We would like to thank the editor, Dr. Corinna Hoose, and two reviewers. Your insightful comments helped improve this paper. A list of our response and the marked-up manuscript are given below. Referees' comments are in black, while our responses are written in Italic.*

**Response to anonymous Referee#1**

General comment
This study examines characteristics of cloud phases in five, frequently occurring, overlapping configurations, over a wide area of southeast Asia. For this purpose, the authors use combined CloudSat-CALIPSO and MODIS data. The different cloud phases are examined in terms of their seasonality and relationship with meteorology, and frequency of occurrence. Their heterogeneity and spectral radiance characteristics are examined in combination with corresponding MODIS data. Associations with MJO and ENSO are also investigated.
The study is to a large extent comprehensive. The results are discussed adequately, and the findings combine verification of previously known characteristics of specific cloud phases/types and their combinations, with new insights over their future usefulness in field campaigns and GCM evaluations. For these reasons, I recommend acceptance of this manuscript for publication in ACP. I include a list of minor comments and technical corrections for the authors to consider.

Comments
Page 5, lines 25-28: it would be useful to report how often these "multi-layer, same phase" vertical structures occur, and discuss possible consequences of this simplification.

*Thanks for this great question.*

*First, the multi-layer, same-phase clouds occur much less frequently than one-layer cloud (see figure below). For example, **for the ice-only clouds**, the total frequency is 28.6% composed by 20% one-layer and 8.6% multi-layer cloud. **For the ice clouds above liquid clouds**, one-layer ice cloud frequency is 13.8% and multi-layer ice cloud frequency is 6.4%. Similarly, for **the liquid-only cloud** (total frequency ~16%), one-layer liquid cloud is 14.0% and multi-layer liquid cloud is only 2%.*

[Figure]

*Fig. 1 cloud phase classification.*

*Second, though the frequency of multi-layer, same phase cloud is relatively small, we have carefully examined the biases due to our simplified classification to cloud distributions, cloud spatial heterogeneity and spectral radiative features. For example, the distributions (Fig. 2) of one-layer and multiple-layer ice clouds above liquid clouds are similar, i.e. relatively large frequency of multiple-ice clouds occurs where frequent one-layer ice clouds occur.*

*Third, to further check the bias to spatial heterogeneity and radiation, we have selected 10-days data in January 2007 to compare the difference between the one-layer and multiple-layer, same phase clouds. Figure 3 shows the results for ice-above-liquid clouds – the multi-layer ice cloud above liquid cloud moves the PDF a little off that of all-cloud, but the bias is small and doesn't change our analysis.*

[Figure]

*Fig.2 Distributions of ice-above-liquid clouds decomposed to two cases: one-layer ice cloud above liquid cloud (first column) and multiple-layer ice clouds (second column) above liquid clouds.*

[Figure]

*Fig. 3 $H_\sigma$ and $R_{0.645}$ PDF for ice-above-liquid clouds: black for one-layer ice cloud above liquid cloud, blue for multi-layer ice cloud above liquid cloud and red for all ice-above-liquid cloud.*

*We have responded by summarizing these finding in Sect. 2.2 (Page8, Lines 40-41 and, Page 8, Lines 1-7 in revised version) and by adding the following statements and with appropriate additions to Table 3:*

*'As stated in Sect. 2.2, we classify clouds according to cloud phase and cloud layer in five main groups: ice-only, liquid-only, mixed-only, ice-above-liquid, ice-above-mixed clouds. Each group contains both single and multiple layers of the same phase. Our analysis (Table 3) shows that one-layer-one-phase clouds have much larger frequency than multi-layer-same-phase clouds. For example, multi-layer ice-only cloud (~8.6%) occurs less frequently than one-layer ice-only cloud (20%). Liquid-only clouds mostly form in a single layer (14%) and the frequency of multi-layer liquid-only cloud is only 2%. A careful comparison between single and multiple layers of the same phase clouds shows no significant difference in the properties that we're interpreting, which justifies our simpler classification.'*

Page 6, lines 24-25: I don't understand how MODIS detects less clear-sky cases than CC by missing some cloudy cases. Shouldn't it be the other way around?

*Yes, you're correct. MODIS detects more clear sky than CC. As displayed in Table 2, MODIS clear sky samples are 4 623 583 (32%), while CC clear sky samples are 2 587 635 (17.9%).*

*Now in Page 6, Lines 20-22, it is corrected as:*

*'We also found that 29% of CC ice-only clouds are reported as clear sky by MODIS, indicating that MODIS misses some thin cirrus in the SEA region – a point also made in Reid et al. (2013).'*

Figure 5: this figure is hard to read. Please consider replacing with 2D plots.

*As suggested, Figure 5 is replaced with 2D plots.*

Page 16, lines 26-27: how is the frequency of occurrence related to the average reflectance? Shouldn't they be thicker to have higher R?

*The averaged reflectance is not weighted by occurrence frequency, and thus you're right that thicker clouds have higher R.*

*Now in Page 13, Lines 30-32, it is modified as:*

*'The averages of the reflectance and the brightness temperature (BT) (not weighted by cloud occurrence frequency) for each cloud group are shown over SEA.'*

Page 18, lines 25-27: it is hard to verify this statement based on Fig. 14d. For example, ice-above-liquid after 07/08 does not agree well.

*Thanks for this comment. We have updated our results using the up-to-date CC data (R05), which show better cloud phase variation with ENSO. To verify this statement, we calculate the correlation coefficient between MODIS ice cloud anomaly and CC ice-only, ice-above-liquid, ice-above-mixed and mixed-only clouds, respectively. Results show that the CC 'ice-contained clouds' are well correlated with MODIS ice clouds as listed in the following table:*

| | Ice-only | Ice-above-liquid | Ice-above-mixed | Mixed-only |
|---|---|---|---|---|
| *Coefficient correlations with MODIS ice cloud* | *0.7* | *0.75* | *0.86* | *0.72* |

*Original statement: 'The MODIS ice cloud anomaly agrees with that of ice-only, ice-above-liquid, ice-above-mixed and mixed-only clouds from the CC observations (Fig. 14d), because most of these CC clouds are reported to be ice by MODIS (Table 2).'*

*Now in Page 17, Lines 28-30, it is revised as:*

*'The MODIS ice cloud anomaly correlates well with that of CC ice-only, ice-above-liquid, ice-above-mixed and mixed-only clouds (Fig. 14d) with correlation coefficients greater than 0.70 (significant at 99% confidence level).*

Page 18, lines 29, 30: what is considered "abnormal" in the heterogeneity index variation?

*We call the positive (negative) anomaly as abnormally high (low). To avoid this confusion, we discard the wording 'abnormal' and rephrase our results accordingly.*

*For example, the original statement "$H_\sigma$ is observed to be abnormally small in La Niña year due to the increase of 'ice-contained clouds' and abnormally large in El Niño year because of decreased 'ice-contained clouds and the expose of liquid clouds",*

*Now in Page 17, Lines 32-34, it is revised as:*

*'$H_\sigma$ anomaly is observed to be negative in La Niña year due to the increase of 'ice-contained clouds' and positive in El Niño year because 'ice-contained clouds' decrease, exposing more liquid clouds.'*

Page 19, line 41: "where are relatively cold". Are you referring to the lower troposphere

conditions? Please clarify.

*Yes, you're right.*

*It is now in Page18, Lines 38-40 corrected as:*

*'Liquid-only clouds appear frequently in winter and spring over southeast China and East China sea where the lower troposphere is relatively cold, dry and stable'*

Figure 6: what are the vertical dashed lines?

*Thanks for pointing out this issue. We now add the explanation in Figure 6 caption: The vertical dashed lines in a)-e) indicate the median values of the PDFs.*

Figure 9: seasonality symbols are not clear. Please consider plotting differently or including a table.

*The values of each symbols are now summarized in Table 5.*

Page 2, line 14: "macrophyscial" should read "macrophysical".

*Corrected*

Page 2, lines 28-31: please consider rephrasing or breaking this long sentence.

*Original statement: "Particularly, cloud radiative effects in the LW are reduced at the top of atmosphere (TOA) for high over low clouds compared to single-layer high clouds and much stronger than single-layer low clouds, which nicely demonstrates the importance of accurately representing cloud vertical structures in GCMs."*

*Now in Page 2, Lines 26-28, it is revised as:*

*'Particularly, the radiative effects at the TOA in the LW of high over low clouds are weaker than high clouds but much stronger than single-layer low clouds. These studies nicely demonstrate the importance of accurately representing cloud vertical structures in GCMs.'*

Page 3, lines 8-9: do you mean "has not yet been examined"?

*Yes, it is corrected.*

Page 9, line 1: "cloud" should read "could".

*Thanks, it is corrected.*

Page 10, line 4: "summaries" should read "summarizes".
*Thanks, it is corrected.*

Page 11, line 22: "CLIPASO" should read "CALIPSO".
*Thanks, it is corrected.*

Page 13, line 3: please omit "that".
*Corrected.*

Page 13, line 42: "spatial" should read "spatially".
*Corrected.*

Page 14, line 2: "it" should read "its".
*Corrected.*

Page 14, line 35: "reflected" should read "reflective".
*Corrected.*

Page 14, lines 42-43: do you mean "refractive index"?
*Thanks for your correction. It is corrected.*

Page 15, line 16: please consider replacing "aware" with e.g. "note that".
*As suggested, we replace 'aware' to 'note that'*

Page 16, line 25: "and thicker" should read "and they are thicker".
*Corrected.*

Page 16, line 33: 0.546 should read 0.645.
*Corrected.*

Page 17, lines 18-20: please rephrase.

*The original statement:*

*'Area with small $H\sigma$ is surrounded by relatively large $H\sigma$ values, indicating the locations where occur suppressed MJO phase associated with more liquid clouds.'*

*Now in Page 16, Lines 16-18, it is revised as:*

*'Areas surrounding the convective center are with relatively large $H_\sigma$ values, indicating that the locations of suppressed MJO phase are associated with more liquid clouds.'*

Page 17, line 22: please consider replacing "are with" with e.g. "display".

*Revised as suggested.*

Page 17, line 24: "connective" should read "convective".

*Corrected.*

Page 19, lines 13-14: please rephrase.

*The original statement, 'As cloud phases vary interannually and hence change the spatial heterogeneity, i.e., being smoother in La Niña year than normal and vice versa in El Niño year.'*

*Now in Page 18, Lines 10-11, it is rephrased as:*

*Overall, the cloud phase varies interannually, as does $H_\sigma$ , i.e., being smoother in La Niña years compared to El Niño years.*

Page 19, line 15: "well correlates" should read "correlates well".
*Corrected.*

Page 20, line 18: "heterogenous" should read "heterogeneous".
*Corrected.*

Page 21, line 5: please replace "attribute" with "contribute".
*Corrected.*

**Detailed response to reviewer #2' comments**

In this paper, the authors investigate cloud properties as seen by different A-train satellites over the Southeast Asia. They further divide clouds into 5 cloud types as a function of their cloud phase and different overlapping possibilities among the cloud-type layers. In the last part of the manuscript, they study possible links between these 5 types and MJO and ENSO conditions. While the topic of this paper aligns with the scope of the journal, it is difficult to judge the novelty of the analysis since most of it is a kind of climatology rather than new results. However, I acknowledge a tremendous amount of work from the authors. Yet, the paper is too long and descriptive, which makes it difficult to follow. In addition to trimming the manuscript, I have a couple more concerns to address before recommending this paper for publication. A more detailed explanation is provided below.

*The paper is indeed a climatological study, focusing on cloud phase characteristics. Since the climatological characteristics of cloud phase have not been addressed, the results are novel. The abstract highlights several of the new key results, and we note that Referee#1 commented on the novelty of our results. The length and descriptive nature of the manuscript is addressed below.*

Main concerns
1) This paper is too long and descriptive. It is hard to follow and I often lost track of the goal of the sections. Every section should be reduced in size and I would recommend focusing on specific findings relevant to the topic of the study rather than describing every subplot of the figures as well as the behavior of each cloud types.
*Thank you for this comment.*
*We have worked to reduce the length of most sections of the manuscript in response to this comment. We have also provided additional edits throughout the manuscript to improve*

*readability. In reducing the text, we did aim to let the figures and tables speak for themselves, but key findings from the figures do need to be discussed. We note that Referee#1 stated "The results are discussed adequately, ..." so we tried to strike a balance between the referees' comments that are at odds with one another.*

2) If I understand correctly there is no filtering of the data whatsoever to make them consistent with each other. I find this a little concerning. For example, in section 3.1.5 when comparing the spatial heterogeneity index with CloudSat-CALIPSO, all pixels are used including those where CC and MODIS cloud masks disagree. This may result in large biases as explained by the authors later on. It would be best to keep in the main analysis the pixels where CC and MODIS agree.

*Thanks for this comment. We did examine this issue (results shown in figure below), and some of these issues are discussed in the paper (as noted by the reviewer), with results on CC and MODIS both being "clear" shown in Table 2. If we forced the analysis to be the same class consistency, then that leaves us vulnerable to carrying MODIS cloud detection and classification errors into our analysis, which we didn't want to do. The point is just to focus on CC classification as a function of $H_\sigma$, without the additional issues that MODIS cloud detection and classification would bring to the interpretation (i.e., discussion as to why the red and black curves look different below is entirely due to MODIS cloud detection and classification limitations—miss some thin or small clouds).*

[Figure]

*Fig. the $H_\sigma$ PDF for clear, ice and liquid cloudy skies: black for CC detections, and red for the samples agreed by both CC and MODIS.*

3) The authors use the version R04 of the 2B-CLDCLASS-LIDAR product. This version is not free of uncertainties in particular when it comes to detecting shallow cumulus clouds. It's been shown that this version overestimates the amount of shallow cumulus clouds (https://www.earth-syst-sci-data.net/11/1745/2019/essd-11-1745-2019.html). Similarly, nothing is said about any kind of uncertainty in the cloud phase retrieval of this product. For example, has this product been evaluated against other cloud phase dataset (ground-based, satellite or in situ?). I know that the cloud phase confidence considerably decreases when the lidar is totally attenuated and the decision tree only relies on the radar signal. I would suggest the authors to mention these at least.

*Thanks for pointing us to the new version 2B-CLDCLASS-LIDAR data (R05), which was not available at the time of our original analysis.*

*1) We have updated all our results using Release 05 (R05) of both 2B-CLDCLASS-LIDAR and 2C-ICE. Compared to the R04 version, R05 shows more ice-only (0.8%↑), mixed-only (0.5%↑) and ice-above-mixed (1.9%↑) clouds, but less liquid-only (0.9%↓) and ice-above-liquid (0.8%↓) clouds in the Southeast Asia region (compare Table 3 in the revised and discussion paper). Also, the new 2B-CLDCLASS-LIDAR product displays a better interannual variations of cloud phase associated with ENSO (see subfigure in Fig. 14d between 07/08 and 07/09). Fortunately, the small changes didn't impact any of our conclusions.*

*2) We agree with the reviewer's concern on the uncertainties of the 2B-CLDCLASS-LIDAR cloud phase retrieval. Characterizing uncertainties in classification does require a truth to compare against, which doesn't exist for cloud phase. When such truths are lacking, the standard approach in validating cloud classification results is to validate the thresholds used in the classification algorithm (Rossow et al., 1989). The thresholds used in the classification algorithm for 2B-CLDCLASS-LIDAR cloud phase is discussed in Section 2.1 and references therein. Still, this doesn't achieve quantitative uncertainty characterization on cloud phase that a comparison to "truth" can give. To date, there are not any evaluation of the CC cloud phase against any other cloud datasets. This is why we performed a comparison of CC and MODIS cloud phase as shown in Table 2 and described in Sect. 2.3 in the paper. Overall, most of CC ice-only, ice-above-liquid, ice-above-mixed and mixed-only clouds are reported to be ice by MODIS, and most of CC liquid-only clouds are also detected to be liquid by MODIS. This comparison allows us to better interpret our results in later sections (e.g., Sect. 3.1.6 and Sect. 3.3).*

*3) We agree that when lidar signal is totally attenuated, confidence level of cloud phase is lowered down. We mention this information in Page 4, Lines 34-35.*
*'When the lidar signal is totally attenuated, the cloud phase is determined only by Ze and temperature, which lowers down the confidence level.'*

4) Finally, the authors consistently mention that their results could be used for model evaluation but fail to explain how. I understand it is tempting to sell any observational result as a possible constraint for model, but if the authors want to do so, they need to explain how and why, which is not done here.

*In the paper (Page20, L12-24), we added the following paragraph to make the model-observation comparison clearer.*

*"Finally, we note that our results may be used to evaluate a model's verisimilitude in capturing cloud properties, particularly phase and spectral characteristics. For example, we show summaries of spectral radiance at the TOA segregated by cloud phase and overlap conditions that can serve as a basis for comparing to those computed from model outputs—a similar approach given by previous research (Hashino et al., 2013; Masunaga et al., 2010; Yao et al., 2020). Since these models also use the plane-parallel assumption in computing the spectral radiation leaving the TOA, careful comparisons between model and observations can use Hσ as*

*a measure of departure from the plane-parallel assumption in a manner similar to* (Loveridge and Davies, 2019), *where they used Hσ within their analysis in examining GCM clouds in different sectors of southern hemisphere cyclones. The use of Hσ also extends its application to gauge biases in other satellite products used in model evaluation (e.g. Gettelman et al., 2015; Song et al., 2018), such as cloud optical depth and effective radius, since biases in these products have been noted to covary with $H_\sigma$(Fu et al., 2019; Zhang et al., 2016)."*

Minor comments
Throughout the manuscript, the authors use plural with the term cloud phase, I feel like most of the time, it would be better to use singular.

*As suggested, we correct the 'cloud phases' to 'cloud phase' in most places.*

P2 L35: improve GCM performance => improve climate simulations

*Corrected.*

P4 L8: in the lower troposphere => below 8 km

*We change the statement as 'in the lower troposphere (i.e. below 8.2 km)'*

P4 L8: in the upper troposphere => above 8 km

*We change the statement as 'in the upper troposphere (i.e. above 8.2 km)'*

P4 L21: Please specify the version. From your table, I believe you use the version R04.

*Data version is added and now all results are updated using R05 data.*

P4 second paragraph: You don't describe how the algorithm works when the lidar signal is completely attenuated. The cloud phase is then based only on Ze and T thresholds, which substantially decreases its confidence level. Since the region under study is dominated by convective clouds, this situation may occur very often.

*Yes, it is true that Ze and T threshold is used to deduce cloud phase in the radar-only region, which could lower down the confidence level.*

*This information is mentioned in Page 4, Lines 34-35 in the revised paper:*

*'When the lidar signal is totally attenuated, the cloud phase is determined only by Ze and temperature, which lowers down the confidence level.'*

P8 L13: I don't understand the unit of LTS, it's supposed to be in K (or ËŽC).

*The unit of LTS depends on how to define it. We define static stability same as Frierson and Davis, (2011) and Li et al. (2014), i.e., $\frac{\partial \theta}{\partial z}$, which has the unit of K/km, while in some other studies such as Klein and Hartmann, (1993), they used the definition of $\Delta\theta = \theta(p = 700mb) - \theta(p = sea\ level\ pressure)$, whose unit is K.*

*To avoid the confusion, in Page 8, Lines 1-3, we have revised the text as:*

*'The lower-troposphere static stability (LTSS=$(\theta_{z=3\ km} - \theta_{z=0})/3\ km$) and the upper-troposphere static stability (UTSS = $(\theta_{z=tropopause} - \theta_{z=tropopause-3\ km})/3\ km$) are shown in Figs 2d1-d4, where the $\theta$ is potential temperature in unit of K.'*

P8 L18-23: The CLDCLASS-LIDAR product provides a cloud fraction (between 0 and 1) per layer so how do you get cloud and sample numbers?

*Whenever the lidar cloud fraction within a radar volume is reported to be greater than zero, we count that radar sample as cloudy. The frequency reported in Figure 3 is the frequency of these samples.*

*To clarify this information, we now add the text in Page 4, Lines 17-18:*

*'This product reports the lidar cloud fraction that records how many lidar profiles are contained in a radar resolution'*

*and in Page 4, Lines 39-40:*

*'Four years of 2B-CLDCLASS-LIDAR data, version P1_R05 (2007-2010) with lidar cloud fraction greater than zero are used.'*

P8 L25: Large => large
*Corrected.*

Fig. 3: why do you use this particular cross section rather than a zonal mean.

*We have updated the results using the zonal mean.*

P8 L41: "Also: : : of cirrus". Why do you mention MISR here out of the blue? Also this sentence is confusing.

*We have deleted these statements to get rid of the confusion.*

P9 L1: Can you elaborate on this statement? DO you mean for that region?

*The original statement: " Low-level clouds cloud have a high chance to be covered by the upper ubiquitous ice clouds (Yuan and Oreopoulos, 2013), which is further quantified in next section."*

*To make this clear, it has been revised in Page 8, Lines 36-38,*

*'As shown in Yuan and Oreopoulos, (2013), low-level clouds have a high chance to be overlapped by upper clouds in the warm pool region. In the next section, we will examine cloud overlap with a focus on cloud phase.'*

P9 L19: Confusing sentence.

*The original statement: "However, liquid-only clouds have very small frequencies (< 10%) between 10°S-10°N where widely distribute ice clouds, which indicates that liquid clouds occurring here are likely being covered by ice clouds, hence, they are grouped as ice-above-liquid cloud class."*

*Now in Page 9, Lines 19-21 , it is rephrased to make in clearer:*

*'Elsewhere, liquid-only clouds have very small frequencies (< 10%). The annual mean frequency of liquid-only cloud is ~16.0%.'*

P11 L13: This should appear in the data section along with the other uncertainties related to the datasets.

*As suggested, we mention these uncertainties due to instrument limitations in Sect. 2.1 (Page 4, Lines 35-37), where we describe the 2B-CLDCLASS-LIDAR data.*

*'Also, in cases of thick ice clouds attenuating lidar signals over shallow liquid clouds that are missed by the radar, only ice clouds are reported in the profiles. Biases due to instrument limitations are kept in mind in our analysis.'*

P11 L22: CLIPASO => CALIPSO
*Corrected.*

P11 L22: attenuated by clouds with optical thickness greater than 3.
*It is revised as suggested.*

P12 L14: spatial => spatially
*Corrected.*

P12 L26: "due to the small optical thickness of the.."
*It is revised as suggested.*

P12 L27: It's unclear to me why the authors constantly refer to MISR for no reasons since MISR observations are used in this study.
*Thanks for pointing out this. In the new version, we remove the contents related to MISR to avoid the confusion.*

P12 L36: This sentence needs re-wording.

*Original statements: "While these clouds are locally homogenous, hence favoring the plane-parallel assumption in radiation computation (Ham et al., 2015)."*

*Now in Page 12, Lines 18-19, it is revised as:*

*'These clouds are locally homogeneous and hence favor the plane-parallel assumption in radiation computation.'*

P13 L1-2: I would strongly recommend excluding pixels in which CC and MODIS disagree in the main figures rather than only mentioning it as "not shown".

*Comment addressed earlier.*

P13 L2-4: There is no main verb in this sentence, please re-word.
Also, are you referring to shallow cumulus clouds? In this case it would be rather easy to validate your hypothesis by focusing on a shallow Cu dominated region, such as the Barbados. However, depending on the MODIS product used, Pincus et al 2012 reported that a substantial amount of these clouds (partially-filled pixels) are excluded of the cloud product. Another important thing to note is that, R04 over-estimate shallow Cu cloud fraction (see main concern comments).

*1) Original statement:*

*"Indeed, many small liquid clouds with size ranging in a few tens to hundreds of meters (e.g., Koren et al., 2008) that are difficult to be measured by MODIS as reported in Zhao and Di Girolamo (2006)."*

*Now in Page 12, Lines 27-29*

*'Indeed, many small liquid clouds with size ranging in a few tens to hundreds of meters can go undetected by MODIS (Zhao and Di Girolamo 2006).'*

*2) Here, we are referring to small and shallow cumulus clouds undetected by MODIS--with their sizes smaller than 1 km. Pixels containing these small clouds could be reported to be clear by both CC and MODIS due to their relatively large spatial resolutions. To validate our hypothesis, we revisit the MODIS and Advanced Space Thermal Emission and Reflection Radiometer (ASTER) data (15 m resolution) used in Zhao and Di Girolamo (2006) over the tropical western Atlantic (Rain in Cumulus over the Ocean field campaign, near Barbados – check Fig. 1 in Rauber et al., (2007)). By excluding the MODIS clear sky pixels that contain ASTER reported clouds, i.e. a focus on MODIS-ASTER clear sky pixels, the long tail of the $H_\sigma$ is strongly reduced (see the Cyan line in the following figure). This is consistent with our hypothesis that small liquid clouds contribute to the long tail of $H_\sigma$ PDF.*

*To make this clear, now in Page 12, Lines 29-35, we have included the statements:*

*'We revisit the MODIS and Advanced Space Thermal Emission and Reflection Radiometer (ASTER) data (15-m resolution) used in Zhao and Di Girolamo (2006) over the tropical western*

*Atlantic. The long tail of Hσ PDF is significantly reduced, i.e. frequency change from 0.3% to 0.1% at Hσ ~ 0.1, when the ASTER data is applied to exclude the MODIS clear sky pixels that contain ASTER reported clouds. This further affirms that the undetected clouds in MODIS and CC clear sky pixels contribute to large Hσ values, which may impact at least 20% clear-sky samples whose Hσ > 0.1 (Figs. 7a,b)'*

*3) In terms of the concern on R04 data, we have updated our results with the R05 version data.*

[Figure]

*Fig. $H_\sigma$ PDF for clear skies obtained from CC and Aqua MODIS over Southeast Asia and from Terra MODIS and ASTER-Terra MODIS over the tropical western Atlantic region.*

P13 L33: Proves seems a bit strong.
*We replace 'proves' as 'indicates'*

P14 L35: reflecting
*It is revised as 'reflective'.*

P17: Why are you showing MJO phases? It's been documented in many many studies already. What does this bring to your study?

*We agree that MJO is well documented in different aspects, including the related cloud type, radiative, dynamic and thermal dynamic characteristics. However, the cloud phase and the corresponding heterogeneity are less studied.*

*To make it clear, Page 15, Lines 40-43, we added the following statements:*
*'This section discusses the features of cloud phase associated with the intraseasonal 30-90 day MJO. Previous studies have provided full overviews of the radiative (in terms of OLR), dynamic and thermal dynamic characteristics of the MJO (Knutson et al., 1986; Riley et al., 2011; Wheeler and Hendon, 2004; Zhang, 2005). The purpose of this study is to focus on how the cloud phase characteristics discussed in previous sections vary with MJO phases.'*

P17 L38: "suppressed"
*It is corrected.*

P18 L3: I don't understand the meaning of this sentence and I don't see how this could be used to validate climate models.

*Original statement: 'Overall, the eastward-propagating $H_\sigma$ patterns-a behavior similar to OLR pattern (e.g., Wheeler and Hendon, 2004), indicate that the $H_\sigma$ could be useful for MJO studies such as serving as an observed-based parameter that are sensitive to cloud phase, to track MJO position and validate MJO simulations in climate models.'*

*Now in Page 17, Lines 4-6, it is revised as:*
*'The eastward-propagating $H_\sigma$ patterns vary with MJO, indicating that $H_\sigma$ could be useful for MJO studies, such as serving as an observed-based parameter to track the MJO position.'*

P18 L8-20: It basically shows there are more convective clouds.

*We remove the reflectance and brightness temperature from Fig. 13 and rewrite the whole paragraph to emphasize the heterogeneity variations (see Page 17, Lines 17-23).*

P19 L13-14: Another sentence without meaning, please re-word.

*Original statement:*

*'As cloud phases vary interannually and hence change the spatial heterogeneity, i.e., being smoother in La Niña year than normal and vice versa in El Niño year.'*

*In Page 18, Lines 10-11, It is revised as:*

*'the cloud phase varies interannually, as does $H_\sigma$ , i.e., being smoother in La Niña years compared to El Niño years.'*

P19 L36: "preferentially occur in"
*It is revised as suggested.*

P19 L38: occur => are
*It is revised as suggested.*

P21 L10: a comma is missing after ENSO L11.

*It is corrected.*

Overall, I would suggest rephrasing this sentence because I don't think the authors can claim the heterogeneity index captures MJO or ENSO. At best, it varies for the different ENSO/MJO phases, but it's definitely not well correlated. For the second part of the sentence, unless the authors explain how one could use this for model evaluation, I'd recommend to remove.

*Original statement: "The observed $H_\sigma$ values capture the MJO and ENSO features, implying that the Hσ is able to track MJO and ENSO and provides a way to validate their simulations in GCMs"*

*In Page 20, Line 6-7, the following sentence replaces the original statement:*

*'The observed Hσ varies with the ENSO index with a correlation coefficient of 0.49 (significant at confidence level 0.99).'*

P21 L14-16: Here again, there is no tool to compare this to models, at least to the best of my knowledge, so unless the authors elaborate on this statement, they should remove this statement. I can envision a qualitative comparison of heterogeneity at best.

*Addressed in point 4 above.*

In this section, We first display theThe spatial distributions of all-sky spatial heterogeneity ($H_\sigma$) anomaly is shown in( Fig. 13) the ENSO index shown in Fig. 14a is used to identify the warm (El Niño) and cold (La Niña) phase of ENSO. Anomalies of clouds, radiation and heterogeneity in El Niño and La Niña years are calculated based on the ENSO index and MODIS data from 2003-2017. The spatial distributions of theses anomalies over SEA are shown in Fig. 13 and their time series are displayed in Fig. 14. As expected, the spatial distributions show negative $R_{0.645}$ anomaly (red) in El Niño year (Figs. 13a1-a4) and positive anomaly (blue) in La Niña year (Figs. 13b1-b4). Opposite patterns are obtained for the $BT_{11}$ (Figs. 13c, d) andGenerally, all-sky $H_\sigma$ anomaly is negative in La Ni-ña year, indicating (Figs. 13e, f). The $H_\sigma$ anomaly indicates that that more ice and mixed clouds (corresponding to positive $R_{0.645}$ and negative $BT_{11}$ anomaly) occurring in La Niña year causinge the spatial heterogeneity to be more homogeneous than normal and vice versa in El Niño year. Also, the anomalies over the Maritime continent tend to be stronger in winter and spring than in summer and fall (Fig. 13). This demonstrates that the $H_\sigma$ varies with ENSO.

Figure 14 displays the time series of monthly anomaly of clouds, radiances and spatial heterogeneity for the areas indicated by the red-dashed box in Fig. 13a1 where are an area sensitive to ENSO signals. As displayed, ice clouds detected by MODIS show similar variations as MODIS all clouds with their anomalies ranging from -0.2-0.2, being positive in La Niña years and negative in El Niño years (Fig. 14b). The MODIS ice cloud anomaly correlates wellagrees with that of CC ice-only, ice-above-liquid, ice-above-mixed and mixed-only clouds from the CC observations (Fig. 14d), as thewith correlation coefficients greater than 0.70 (significant at 99% confidence level). ), because most of these CC clouds are reported to be ice by MODIS (Table 2). Conversely, the anomaly of liquid cloud occurrence is positive abnormally high (negativelow) in El Niño (La Niña) year based on both the MODIS and CC data (Figs. 14b, d). Correspondingly, $H_\sigma$ anomaliesy is are observed to be negative abnormally small in La Niña year due to the increase of 'ice-contained clouds' and positive abnormally large in El Niño years because of the decreased 'ice-contained clouds' decrease, and the expose ofexposing more liquid clouds expose (Figs. 14b, d). Moreover, the correlation coefficient between $H_\sigma$ and ENSO index is about 0.49 (significant at 99% confidence level), further indicating that the change of spatial

heterogeneity is associated with ENSO.

Note that negative (positive) anomaly of  CC liquid-only or MODIS liquid cloud associated with La Niña (El Niño) phase does not mean that total liquid clouds occur less (more) in La Niña (El Niño) year.  The overlying clouds can conceal liquid clouds to be observed from space by passive sensors or by lidar if the overlying clouds are optically thicker than 3. Unlike CC liquid-only or MODIS liquid clouds, the CC ice-above-liquid clouds occur abnormally high in La Niña year (Fig. 14d). When adding up the frequency of liquid-only and ice-above-liquid clouds (i.e. total CC liquid cloud frequency), the anomaly  shows  relationship with ENSO less evident than the liquid-only cloud (the subfigure in Fig. 14d). For example, through the La Niña phase in 2007 winter through 2008 spring,  the anomaly of total CC liquid cloud occurrence is close to be zero, blurring its relationship with ENSO. In Park and Leovy (2004), they showed negative anomalies of low-level clouds during positive ENSO phase using ship observations reported by the Extended Edited Cloud Report Archive (EECRA), i.e. less low-level clouds in El Niño year. While in our study, it is likely that MODIS data and four-year CC data is insufficient to support the relationship between liquid cloud occurrence and ENSO over SEA.

Overall, the ~~interannual variations of clouds over SEA are primarily due to the change of ice clouds, which are consistent with the statements in previous studies that in El Niño year, anomalous subsidence over the Western equatorial Pacific decreases cloud amount by suppressing deep convection and reducing high clouds (Park and Leovy, 2004; Wang and Su, 2015). As a result, the observed radiation at the TOA adjust accordingly (Loeb et al., 2012). Assvaryand hence changethe spatial heterogeneity at the TOA change correspondinglysthan normal and vice versa insieswelly~~ies 
[revised manuscript text omitted]
 microphysical properties associated with the MJO and ENSO from the space-time variability of biases in passive retrievals of these cloud properties caused by departures from the plane-parallel assumption. values capture the MJO and ENSO features, implying that the $H_\sigma$ is able to track MJO and ENSO and provides a way to validate their simulations in GCMs.~~

As the CC observations are sensitive to a wide range of cloud scenarios, cloud patterns shown here can help interpreting the cloud product variability in SEA as discussed in Reid et al., (2013). The results also provide summaries of spectral radiation at the TOA useful for accessing model output of spectral radiance computed from modeled clear and cloud properties, such as those found within GCM or reanalysis products , providingClimatological characteristics of cloud phases provided in this study can serve as a benchmark to improve the performance of climate models in validating their simulations of cloud phases and their vertical overlap, their spatial heterogeneity and spectral signaturess, including cloud. Since these models also use the plane-parallel assumption in computing the spectral radiation leaving the TOA, careful comparisons between model and observations can use $H_\sigma$Hs as a measure of departure from the plane-parallel assumption. Particularly, spatial heterogeneity, a direct measured variable from satellite that reveals subpixel variability of different cloud phases, is not only able to track the MJO and ENSO but is also useful to accessing the performance of GCMs in capturing cloud property variations associated with the MJO and ENSO can use the space-time behavior of $H_\sigma$Hs shown here in a manner similar toevaluate how well GCMs capture subpixel cloudsLoveridge and Davies (2019), where they used $H_\sigma$Hs 
[revised manuscript text omitted]

---

## Author Response (AR2)

We would like to thank the reviewer again for his/her comments. Our response is shown in blue color.

This is the second round of review.
First I want to remind the authors that as a reviewer, I'm required to give my opinion on certain criteria, including the novelty of a study and whether its length is appropriate or not, which I'm doing here based on my (imperfect) knowledge of the literature and regardless of what other reviewers may say. However, it doesn't mean that I find the study irrelevant or not good.

That being said, the authors addressed most of the points I rose in the first round of review but some concerns remain and need to be addressed before I recommend the paper for publication. These are listed below with additional comments regarding some author arguments.

Comments that do not need to be further addressed
1) Novelty of the study
I just want to remind the author that previous literature already largely investigated the cloud phase characteristics globally (e.g., Cesana et al., 2015; Hu et al., 2010; Li et al., 2017; Matus and L'Ecuyer, 2017; Yoshida et al., 2012). Some of that literature also studies the link with radiation for different overlap using the very same product as the authors (e.g., Matus and L'Ecuyer, 2017). Focusing on a specific region doesn't make a study novel, but again it doesn't mean it's not worth being published. However, I acknowledge that the spatial heterogeneity component of the study ¬¬–and its link with cloud phase– is quite new and interesting.

We thank the reviewer for this comment. We are also aware that a series of research has focused on the characteristics of cloud phase (with no overlap information), cloud overlap (with no cloud phase information) and their linkage with radiation (largely broadband not spectral). Paragraph 2 of Introduction reviews this literature and states our motivation to investigate the characteristics of cloud phase overlap and its link to spectral and spatial heterogeneity signatures.

Many references mentioned here are already in the paper. We add Cesana et al. 2015, and Yoshida et al. 2012 to Introduction section to make the literature review more complete. Thank you.

2) Length of the study
While I appreciate the author efforts to shorten the manuscript, I still find the study quite long, but it doesn't bar it from being published.

We thank the reviewer.

Concerns that need to be addressed
1) Uncertainties and caveats related to the observational product
- The authors now better mention the caveats and uncertainties of the product, which is good, in particular the reduction in confidence of the diagnostic when the lidar is completely attenuated. However, they fail to mention how often this happens in their study, a breakdown depending on the category would be helpful (i.e., how often the diagnostic relies on radar only by category).

This region is dominated by deep convection and therefore I would expect most of the observations to be radar only, which is why it has to be quantified, it's essential information for the reader.

Quantifying how often the cloud classification relied on radar-only signals is helpful in certain studies, but not so here. First, it is true that deep convection is active in this region. But, the most common cloud type is cirrus, some of which are spawned by deep convective clouds. Table 3 shows that the combination of ice-only, ice-over-liquid, and ice-over-mixed represent ~71% of all cloud categories. These ice-containing categories occur predominately for cirrus with optical depth < 3.0 (Figure 6), as they require lidar beam penetration for classification. For mixed-clouds, which includes deep convective clouds, the detection of ice at the top with the aid of the lidar is sufficient for classification (even though the lidar beam gets completely attenuated), since the 2B-CLDCLASS-LIDAR only provides the information for cloud layers, not for each radar range gate – so, deep convective clouds are classified as mixed-phase cloud, though there could exist only water at the bottom of the cloud (as discussed in Section 2.1). This last point may well be why the 2B-CLDCLASS-LIDAR product doesn't archive the information when lidar/radar signal is available.

- The authors say "the most comprehensive cloud phase and overlap information to date" p4 L37 I disagree with that statement. There is no paper that supports this statement to the best of my knowledge. The authors themselves stated that they are not aware of any kind of validation of the 2B-CLDCLASS-LIDAR product, which makes it difficult to conclude on whether this product is the most comprehensive to date. There are at least 2 other LIDAR-RADAR cloud phase datasets out there using different methods (DARDAR and Kyushu University products) as well as 3 lidar-only cloud phase products (CALIPSO-ST, GOCCP and Kyushu University), some of which have been validated against ground-based or in-situ measurements contrary to 2B-CLDCLASS-LIDAR product. Please, rephrase.

We thank the reviewer for this comment. What we want to express is that the combined radar-lidar measurements are able to provide comprehensive cloud information including cloud phase and cloud overlap.

The original statement is now rephrased as:
Despite these limitations, the combined radar-lidar measurements provide comprehensive cloud phase and overlap information.

- Finally, the method used by the authors to account for the cloud fraction (i.e., cloudy profile each time the lidar cloud fraction within the cloudsat volume is greater than 0) leads to an overestimate of the cloud fraction in regions of fractionated clouds such as the trade winds and this should be explicitly mentioned in the manuscript (e.g., Cesana et al., 2019; Marchand et al., 2010).

We do not calculate the cloud fraction, such as in Marchand et al. (2010). Instead, we calculate the occurrence frequency of the radar column containing some cloud, even if not fully cloudy. When the lidar cloud fraction is greater than zero, we count the radar column as containing some

cloud. A threshold similar to Cesana et al. 2019 is not adopted since we wanted to include small-size clouds detected by the lidar in our analysis (e.g., Section 3.1.5) and avoid issues with traditional resolution and thresholding effects on cloud fraction as discussed in Marchand et al. (2010) and many of the references we cite.

In P4 Line 39-40:
We revised the statement by adding 'in order to include small size clouds':
Four years of 2B-CLDCLASS-LIDAR data, version P1_R05 (2007-2010) with lidar cloud fraction greater than zero are used in order to include small size clouds.

2) Climate model evaluation argument
I appreciate the effort of the authors to clarify how to use their results to inform model simulations. However, I'm still not convinced by their explanation. The 2B-CLDCLASS-LIDAR cloud ice and liquid frequency cannot be used to evaluate climate models. There are no cloud ice or liquid frequency in the models to compare with. Also, if such diagnostic was available, it would still be not consistent to directly compare the observations with the models without using a method that takes into account the inherent biases of the instruments. For example, one should use a forward simulator that reproduce the 2B-CLDCLASS-LIDAR product process and biases to compare with the models (see for example Masunaga et al., 2010 and Hashino et al., 2013 referenced by the authors, and many other not referenced here). Such simulator doesn't exist.
Additionally, a quick look at Loveridge and Davies –referenced by the authors as an example of how to use their heterogeneity index for model evaluation– also shows that they use a simulator in their study to reproduce MISR and MODIS quantities, then compute their Hindex and evaluate the heterogeneity parametrization. A GCM grid box is typically on the order of hundreds of kilometers with the most recent one being on the tens of kilometers, which is still far larger than the 1km pixel size used in MODIS observations. This is why it can't be used directly to evaluate a GCM (although not true for finer scale models).
I understand how it could be useful for observations as explained in the paper, but in its actual state, these observations cannot be used for pure model evaluation. Therefore, I'm still recommending to remove these statements of the manuscript.

We appreciate the reviewer's concerns. However, satellite cloud product summaries have been used to gauge model performance for more than 40 years. True, there are many issues in doing so, but the practice continues. The use of forward model simulators, in part, attempts to address some, but not all, of the issues. Just because a forward simulator for the 2B-CLDCLASS-LIDAR product doesn't exist, it doesn't mean that someone won't build one in the future. Moreover, our paragraph emphasizes the forward simulation of radiances form MODIS as the basis of comparison.

With the reference to Loveridge and Davies work, they did not use $H_\sigma$ for direct evaluation of any quantities in the model. Instead, the $H_\sigma$ was used to interpret the remote sensing data, arguing that differences in cloud optical depth (for example) between satellite and model should be considered in context of subpixel heterogeneity information such as $H_\sigma$. So $H_\sigma$ is used as part of the model evaluation process (not direct comparison) to identify high confidence observations that act as strong constraints on the model. This is also the point that we emphasize: ' careful

comparisons between model and observations can use Hσ as a measure of departure from the plane-parallel assumption in a manner similar to (Loveridge and Davies, 2019), where they used Hσ within their analysis in examining GCM clouds in different sectors of southern hemisphere cyclones.'

While we disagree with the reviewer and are confident that our statement is on target, we have modified our original statements as (P20 L13-19):

Finally, careful comparisons between model and observations can use Hσ as a measure of departure from the plane-parallel assumption in a manner similar to (Loveridge and Davies, 2019), where they used Hσ within their analysis in examining GCM clouds in different sectors of southern hemisphere cyclones. Hσ can also be used to gauge biases in other satellite products that are used in model evaluation (e.g. Gettelman et al., 2015; Song et al., 2018), such as cloud optical depth and effective radius, whose biases have been noted to covary with $H_\sigma$ (Fu et al., 2019; Zhang et al., 2016).

References
Cesana, G., Waliser, D. E., Jiang, X., & Li, J. L. F., (2015), Multimodel evaluation of cloud phase transition using satellite and reanalysis data, Journal of Geophysical Research, 120(15), 7871–7892. https://doi.org/10.1002/2014JD022932
Cesana, G., Del Genio, A. D., & Chepfer, H., (2019), The Cumulus And Stratocumulus CloudSat-CALIPSO Dataset (CASCCAD), Earth System Science Data Discussions, 2667637(November), 1–33. https://doi.org/10.5194/essd-2019-73
Li, J., Lv, Q., Zhang, M., Wang, T., Kawamoto, K., Chen, S., and Zhang, B.: Effects of atmospheric dynamics and aerosols on the fraction of supercooled water clouds, Atmos. Chem. Phys., 17, 1847–1863, https://doi.org/10.5194/acp-17-1847-2017, 2017.
Hu, Y., Rodier, S., Xu, K. M., Sun, W., Huang, J., Lin, B., et al., (2010), Occurrence, liquid water content, and fraction of supercooled water clouds from combined CALIOP/IIR/MODIS measurements, Journal of Geophysical Research Atmospheres, 115(19), 1–13. https://doi.org/10.1029/2009JD012384
Marchand, R., Ackerman, T., Smyth, M., & Rossow, W. B., (2010), A review of cloud top height and optical depth histograms from MISR, ISCCP, and MODIS, Journal of Geophysical Research Atmospheres, 115(16), 1–25. https://doi.org/10.1029/2009JD013422
Matus, A. V., & L'Ecuyer, T. S., (2017), The role of cloud phase in Earth's radiation budget, Journal of Geophysical Research, 122(5), 2559–2578. https://doi.org/10.1002/2016JD025951
Yoshida, R., Okamoto, H., Hagihara, Y., & Ishimoto, H., (2010), Global analysis of cloud phase and ice crystal orientation from Cloud-Aerosol Lidar and Infrared Pathfinder Satellite Observation (CALIPSO) data using attenuated backscattering and depolarization ratio, Journal of Geophysical Research Atmospheres, 115(16), 1–12. https://doi.org/10.1029/2009JD012334

Koren, I., Oreopoulos, L., Feingold, G., Remer, L. A. and Altaratz, O.: How small is a small cloud?, Atmos. Chem. Phys., 8, 3855–3864, doi:10.5194/acp-8-3855-2008, 2008.